

# OpenBench: a land models evaluation system

Zhongwang Wei[1*], Qingchen Xu[1], Fan Bai[1], Xionghui Xu[1], Zixin Wei[1], Wenzong Dong[1], Hongbin Liang[1], Nan Wei[1], Xingjie Lu[1], Lu Li[1], Shupeng Zhang[1], Hua Yuan[1], Laibao Liu[2,3], Yongjiu Dai[1]

[1]Southern Marine Science and Engineering Guangdong Laboratory (Zhuhai), School of Atmospheric Sciences, Sun Yat-sen University, Guangzhou, China
[2]Department of Geography, The University of Hong Kong, Hong Kong, China
[3]Institute for Climate and Carbon Neutrality, The University of Hong Kong, Hong Kong, China

*Correspondence to*: Zhongwang Wei (weizhw6@mail.sysu.edu.cn)

**Abstract.** Recent Land surface models (LSMs) have evolved significantly in complexity and resolution, requiring comprehensive evaluation systems to assess their performance. This paper introduces The **Open** Source Land Surface Model **Bench**marking System (**OpenBench**), an open-source, cross-platform benchmarking system designed to evaluate the state-of-the-art LSMs. OpenBench addresses significant limitations in the current evaluation frameworks by integrating processes that encompass human activities, facilitating arbitrary spatiotemporal resolutions, and offering comprehensive visualization capabilities. The system utilizes various metrics and normalized scoring indices, enabling a comprehensive evaluation of different aspects of model performance. Key features include automation for managing multiple reference datasets, advanced data processing capabilities, and support for station-based and gridded data evaluations. By examining case studies on river discharge, urban heat flux, and agricultural modeling, we illustrate OpenBench's ability to identify the strengths and limitations of models across different spatiotemporal scales and processes. The system's modular architecture enables seamless integration of new models, variables, and evaluation metrics, ensuring adaptability to emerging research needs. OpenBench provides the research community with a standardized, extensible framework for model assessment and improvement. Its comprehensive evaluation capabilities and efficient computational architecture make it a valuable tool for both model development and operational applications in various fields.

## 1 Introduction

Land surface models (LSMs) simulate the complex interactions among the land surface, planetary boundary layer, rivers and lakes, glaciers and frozen soils, plant physiology and ecology, vegetation dynamics, biogeochemistry, human activities, and other processes occurring on the land surface (Blyth et al., 2021; Dai et al., 2003; Lawrence et al., 2019; Pokhrel et al., 2016). These models play an important role in understanding and predicting various changes in the earth system, serving as a bridge connecting the land surface, ocean, and atmosphere (Fisher and Koven, 2020; Ward et al., 2020; Liu et al., 2024). As such, they are key components of earth system models (ESMs) and have significant impacts on our ability to comprehend and predict weather, climate, hydrological cycles, carbon cycles, and various other environmental factors. In recent decades, LSMs have undergone rapid development, evolving from basic "bucket" models (Manabe, 1969) to advanced multi-module systems (Blyth



et al., 2021) that incorporate both biogeochemical processes (e.g., greenhouse gas, carbon, nitrogen, and phosphorus cycles),
geophysical processes (including land use changes, three-dimensional surface water, subsurface flow, and flooding), as well
as human activities (such as agriculture, reservoir management, and urban development). This evolution has been driven by
advances in hydrology, meteorology, computer, and measurement technology, leading to the development of increasingly
complex models. Concurrent with the increasing complexity of processes represented in LSMs, there has been a significant
improvement in spatial resolution as well. Models have progressed from traditional forecasting scales of 25-100 km (Dai et
al., 2003) to current fine scales of 0.1-10 km (Chen et al., 2024). The increasing complexity and resolution of models require
comprehensive evaluation and analysis of simulation results.

In recent years, various model benchmarking systems (See **Table 1**) have been developed. These systems assess model
performance in comparison to multiple sources of reference datasets. Most of these benchmarking systems consist of
benchmark datasets, evaluation software, metrics, model operating environments, and auxiliary tools. The benchmark datasets
standardize observation, reanalysis, and remote sensing data to evaluate model accuracy in simulating land processes.
Evaluation software includes metrics, execution environments, and tools designed for automated assessment and quantifying
LSMs performance. The operating environment comprises the software and hardware for running evaluations, while ancillary
tools support benchmarking. Despite the importance of LSM evaluation and the development of various benchmarking systems,
several limitations persist in current evaluation approaches. These limitations have become increasingly apparent as the
complexity and resolution of LSMs have increased. One significant area for improvement is the scope of evaluation variables
in most existing evaluation systems. These systems typically focus on some range of commonly used variables, such as water,
heat and carbon fluxes, temperature, and vegetation coverage. This restricted scope fails to capture the full range of processes
simulated by modern LSMs. For instance, TraceMe (Zhou et al., 2021) is primarily designed to evaluate model outputs related
to the carbon cycle, while the MetEva software developed by the National Meteorological Center of China
(https://github.com/nmcdev/meteva) focuses on meteorological fields. However, neither of these tools provides a
comprehensive assessment of land surface processes, nor can they easily adapt to new evaluation indicators or datasets. In
particular, there is a lack of comprehensive evaluation for hydrological cycles and human activities, making it challenging to
assess model performance in these critical areas fully. Human activity, while an important factor affecting surface processes,
is one of the most challenging aspects to model and evaluate. This is primarily due to the small-scale nature of human activity
data (e.g., crop yields, dam operation, and anthropogenic heat) and the involvement of complex socio-economic and land-use
change data. These datasets are often multi-source, complex, and highly uncertain. To date, no evaluation system has been
found to integrate the assessment of human activities in LSMs broadly.

Another significant challenge in current LSMs evaluation practices is the difficulty in conducting inter-model comparisons.
This comparative work is a key step in improving model performance and understanding model differences and uncertainties.
However, the lack of a universal and comprehensive evaluation tool presents significant challenges, especially in the context
of high-resolution complex models and evolving underlying datasets. Traditionally, software tools for evaluating LSMs have
often been customized for specific models or datasets. For example, the evaluation tools for The Canadian Land Surface




Scheme (CLASS) and The Community Atmosphere Biosphere Land Exchange model (CABLE), i.e., AMBER (Arora et al., 2023) and benchcab (https://github.com/CABLE-LSM/benchcab), are optimized for their respective outputs. Customized

software designs lead to several issues. Researchers must invest time in learning specific usage methods and data formats for each new model, limiting their ability to try new models and slowing down comparisons. Different tools using different evaluation criteria and formats make it difficult to compare model performance. Evaluation tools like the International Land Model Benchmarking (ILAMB) platform (Collier et al., 2018) require complex data processing, such as converting model outputs to the CMIP standard. This process consumes time and computing resources, increasing the risk of errors and

potentially affecting the reliability of evaluations. In the meantime, some platforms, such as ILAMB and the Land Surface Verification Toolkit (LVT) (Kumar et al., 2012), offer a wide range of assessments for process variables. However, their spatiotemporal resolution is relatively low (typically at a monthly scale and 0.5°). They have limitations in processing data conversion at different scales, making it difficult to perform simulation evaluations at multiple spatiotemporal scales.

Visual analysis capabilities are another area where current evaluation tools often fall short. Many lack visual functions or

produce low-quality visualizations, making it difficult to display evaluation results effectively. For instance, while some can produce graphical diagnostics, the quality is often insufficient to meet publication standards, and it is unable to customize output. Platform compatibility is also a significant issue, as most evaluation tools are designed to run only on Linux. This limits their application on Windows or macOS operating systems, thus restricting their popularity and accessibility.

**Table 1. The software that can be used or partly used for land surface model evaluation. The abbreviations for specific nouns in the table are described below: AMWG: NCAR's CAM Diagnostics Package; CVDP: NCAR's Climate Variability Diagnostics Package; ESMValTool: Earth System Model Evaluation Tool; PMP: PCMDI's Metrics Package; ILAMB: International Land Model Benchmarking System; MDTF: NOAA's Model Diagnostics Task Force Framework; MPAS-Analysis: analysis for MPAS (Model for Prediction Across Scales) components of E3SM Ocean and Sea-ice analysis for E3SM's MPAS components; E3SM Diags v2.7:**

**The E3SM Diagnostics Package; AMBER: Automated Model Benchmarking; PALS: Protocol for the Analysis of Land Surface models; LVT: Land Surface Verification Toolkit; benchcab: Evaluation Tool for the Land Surface Model CABLE; TraceMe: Traceability analysis system for model evaluation; AMET: The Atmospheric Model Evaluation Tool; MET: The Model Evaluation Tools; MVIETool: the Multivariable Integrated Evaluation Tool.**

| Name | Range of application | Arbitrary spatiotemporal resolution | Cross-platform | Reference | Link |
|---|---|---|---|---|---|
| **MetEva** | GRAPES model | No | Yes | NA | https://github.com/nmcdev/meteva |
| **AMWG** (**retired** | CAM | No | No (Linux) | NA | https://www2.cesm.ucar.edu/working_groups/Atmo |



| | | | | | sphere/amwg-diagnostics-package/ |
|---|---|---|---|---|---|
| **CVDP** | CMIP-style | No | No (Linux) | Phillips et al. (2014) | https://www2.cesm.ucar.edu/working-groups/cvcwg/cvdp |
| **ESMVal** | CMIP-style | No | No (Linux) | Weigel et al. (2020) | https://esmvaltool.org/ |
| **PMP** | CMIP-style | No | No (Linux) | Lee et al. (2023) | https://github.com/PCMDI/pcmdi_metrics |
| **MDTF** | Single point, CMIP-style, NCAR and GFDL model | Yes | No (Linux and macOS) | NA | https://mdtf-diagnostics.readthedocs.io/en/latest/ |
| **MPAS-Analysis** | MPAS model | Yes | No (Linux) | NA | https://github.com/MPAS-Dev/MPAS-Analysis |
| **E3SM Diags** | E3SM model, CMIP-style | Yes | No (Linux) | C. Zhang et al. (2022) | https://github.com/E3SM-Project/e3sm_diags |
| **ILAMB** | CMIP-style | No | No (Linux) | Collier et al. (2018) | https://www.ilamb.org/ |
| **AMBER** | CLASSIC, CTEM | Yes | Yes | Arora et al. (2023) | https://cccma.gitlab.io/classic_pages/benchmarking/ |



| PALS | Single point | Yes | Yes | Best et al. (2015) | https://modelevaluation.org/ |
|------|--------------|-----|-----|--------------------|------------------------------|
| LVT | Various | Yes | No (Linux) | Kumar et al. (2012) | https://github.com/NASA-LIS/LISF/tree/master/lvt |
| benchcab | Single point , CABLE model | Yes | Yes | - | https://github.com/CABLE-LSM/benchcab |
| TraceME | CMIP-style | Yes | No (Linux) | Zhou et al. (2021) | http://traceme.org.cn/ |
| AMET | CMAQ model | Yes | No (Linux) | Appel et al. (2011) | https://www.epa.gov/cmaq/atmospheric-model-evaluation-tool |
| MAT | WRF, UFS, SIMA model | Yes | No (Linux) | Jensen et al. (2024) | https://metplus.readthedocs.io/projects/met/en/latest/ |
| MVIETool | CMIP-style | No | No (Linux) | M.-Z. Zhang et al. (2021) | https://github.com/Mengzhuo-Zhang/MVIETool |

To address these challenges and meet the high standard requirements of new-generation LSM verification and evaluation, we have developed OpenBench (The Open Source Land Surface Model Benchmarking System). The core goal of OpenBench is to provide an open-source, fast, efficient, diverse, and accurate evaluation mechanism for high-resolution land-surface model outputs. OpenBench is designed as a universal and high-performance LSM evaluation system, fully written in Python, that realizes functions such as data processing, evaluation method encapsulation, and result analysis visualization. OpenBench

supports cross-platform, including Windows, macOS, and Linux, enhancing its accessibility and usability across different research environments. OpenBench incorporates evaluation metrics and datasets that account for human activities' on land



surface processes, filling a significant gap in current evaluation systems. The system provides a unified and standardized benchmark test method framework, allowing for efficient and comprehensive system validation and evaluation of typical land surface models such as CoLM, CLM, Noah-MP, GLDAS, and JULES, as well as CMIP styled model output. By ensuring the

widespread sharing of evaluation results, OpenBench aims to advance scientific research and operational work in land surface modeling. The system maximizes the use of available observational and reanalysis data through its efficient data management and processing capabilities.

In the following sections of this paper, we will detail the methodology behind OpenBench, including its system architecture, key components, and the benchmark datasets developed for it. We will then present case studies that demonstrate its application

in evaluating and comparing different LSMs or parameterization, highlighting its capabilities in handling high-resolution data and complex processes. Finally, we will discuss the implications of this new evaluation system for the field of land surface modeling and outline future directions for its development and application.

## 2 Overall Structure

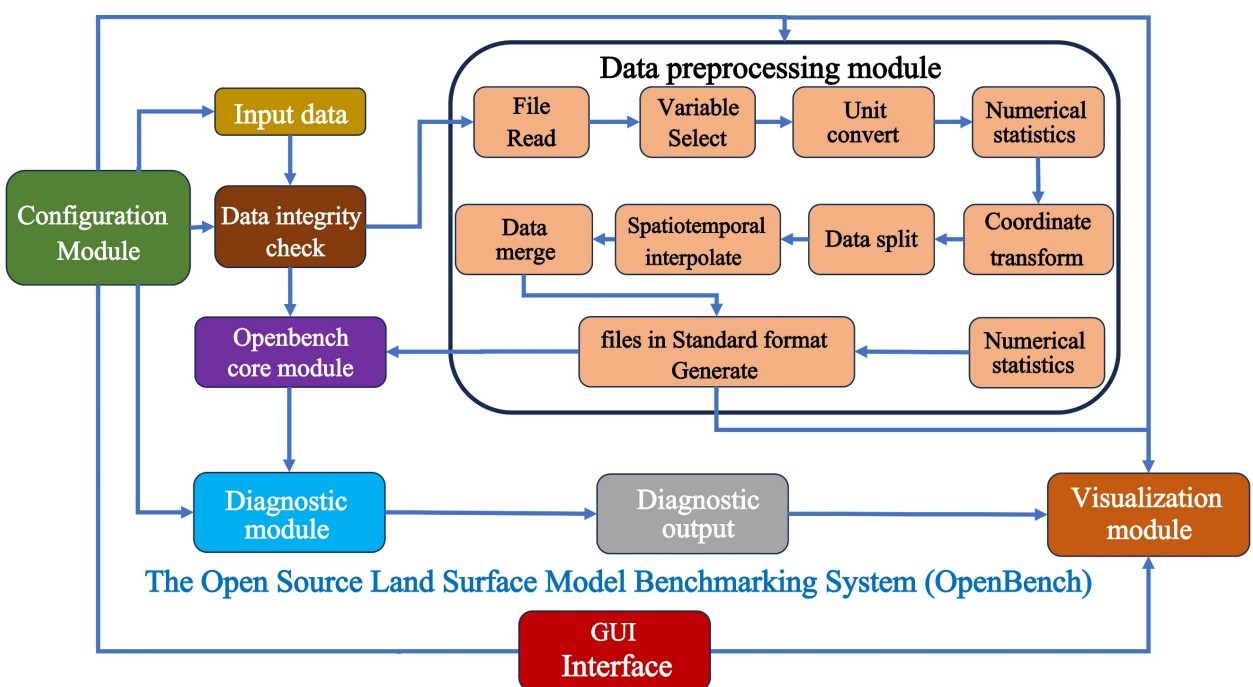

115                                    **Figure 1: General flowchart of the OpenBench**

OpenBench represents a significant advancement in the field of model evaluation and intercomparison. This section outlines the system's overall structure, highlighting its key components and workflow. The benchmarking system is designed with modularity and flexibility in mind, enabling efficient processing of diverse datasets and model outputs. The OpenBench code





is designed to simultaneously handle various data types, including plot scale data (such as station data) and gridded data
(regional or global) for both simulation and reference datasets. The flowchart of OpenBench is shown in **Fig. 1**.

The system includes six components, i.e., configuration management, data processing, evaluation, comparison processing, and
statistical analysis and visualization. The configuration management module utilizes a Fortran namelist-based configuration
approach, allowing users to specify evaluation parameters, data sources, and model outputs. This flexible configuration system
supports easy customization of evaluation scenarios. The data processing module handles the preprocessing of both reference
and simulation data, including temporal and spatial resampling to ensure consistent comparison between datasets with different
spatiotemporal resolutions. The evaluation module implements the core evaluation logic, applying various metrics and scores
to quantify model performance. It supports both gridded and station-based data and adapts its methods accordingly. The
comparison module facilitates multi-model and multi-scenario comparisons, enabling comprehensive analysis across different
models or configurations. Finally, advanced statistical techniques are implemented in the statistical analysis module, providing
deeper insights into model behavior and performance patterns. The system also includes capabilities for generating
visualizations of evaluation results, which are crucial for interpreting and communicating findings provided in the visualization
module.

The system's workflow follows a logical sequence of operations. The process begins with initialization, where command-line
arguments are parsed, and configuration files are read. This stage sets up necessary directories and initializes key variables,
laying the groundwork for subsequent operations. The system then moves into the data preparation phase, where both
observational and model data are processed to ensure compatibility in terms of temporal and spatial resolution. This crucial
step handles various data formats and structures, normalizing them for consistent analysis. At the core of the system is the
evaluation process. Here, a wide array of metrics and scores are applied to quantify the agreement between model outputs and
observational data. This step is highly parallelized to efficiently handle large datasets, allowing for comprehensive assessment
across multiple variables and timeframes. If multiple models or scenarios are being evaluated, the system performs comparative
analyses to highlight relative strengths and weaknesses. This comparison stage provides valuable perspectives into model
performance across different conditions or implementations. Following the primary evaluation, the system conducts advanced
statistical analyses to gain profound understanding from the evaluation results. This may include uncertainty quantification,
trend analysis, or other sophisticated statistical methods. The final stages involve result generation and visualization. The
system compiles evaluation results, generates summary statistics, and prepares data for visualization. The system can produce
various charts, graphs, and maps to effectively communicate the evaluation outcomes. Throughout these stages, the system
demonstrates flexibility in handling different types of data (grid-based or station-based), various temporal resolutions, and a
wide range of environmental variables. It also incorporates specialized handling for different land surface models, recognizing
the unique characteristics and outputs of each. This comprehensive approach allows for a thorough, standardized evaluation
of land surface models, providing valuable feedback for model development and application in Earth system science.

Nevertheless, OpenBench is developed to serve as a specialized tool for land surface model output analysis, evaluation and
comparison. The software package is freely available to the community. The code is modular and can be easily extended or



modified to accommodate specific requirements of different evaluation tasks. OpenBench relies on various popular and well-established Python packages specific to scientific computing stack: NumPy (Harris et al., 2020), Xarray (Hoyer and Hamman, 2017), Pandas (Mckinney, 2010), SciPy (Virtanen et al., 2020), Matplotlib (Hunter, 2007), Cartopy (Met Office, 2010), Dask (Rocklin, 2015), and Joblib (Joblib, 2020). The remap functions have several options: SciPy, Cdo (Schulzweida, 2023) , xesmf (Zhuang et al., 2023), and xarray-regrid (Schilperoort et al., 2024) are available for selection. We use as few packages as possible, reducing dependencies to improve performance and compatibility. The software is developed and hosted on GitHub and is distributed under the Apache-2.0 license. The latest version of OpenBench can be found in the Zenodo repository, where it has been assigned a Digital Object Identifier (https://doi.org/10.5281/zenodo.14540647).

## 3 Evaluation

### 3.1 The metric index

**Table 2. Metrics employed in OpenBench**

| Metric | Full name | Range | Reference | Additional Description |
|--------|-----------|-------|-----------|------------------------|
| **Bias metrics (The smaller is better, ideal value is 0)** | | | | |
| BIAS | Bias | $[-\infty, +\infty]$ | Cole (1981) | - |
| PBIAS | The percentage of bias | $[-\infty, +\infty]$ | Sorooshian et al. (1993) | - |
| APBIAS | Absolute Percent Bias | $[0, +\infty]$ | Sorooshian et al. (1993) | - |
| PC_MAX | Percent Bias of Maximum | $[-\infty, +\infty]$ | X. Zhou et al. (2024) | Measuring the bias of a model when predicting the maximum value. |
| PC_MIN | Percent Bias of Minimum | $[-\infty, +\infty]$ | X. Zhou et al. (2024) | Measuring the bias of a model when predicting the minimum value. |
| PC_AMPLI | Percent Bias of Amplitude | $[-\infty, +\infty]$ | X. Zhou et al. (2024) | Measuring the bias of a model when predicting the data range. |
| APFB | Annual high flow percentage bias | $[-\infty, +\infty]$ | Mizukami et al. (2019) | Measuring the relative bias between simulated and observed annual peak flows |
| PBIAS_HF | Percent Bias of High Flows | $[-\infty, +\infty]$ | Mizukami et al. (2019) | Measuring the model's bias in predicting high flows |





| | | | | (typically above the 98th percentile). |
|---|---|---|---|---|
| PBIAS_LF | Percent Bias of Low Flows | [-∞, +∞] | Mizukami et al. (2019) | Measuring the model's bias in predicting low flows (typically below the 30th percentile). |
| PBIAS_FDC | Percent Bias in the Slope of the Mid-segment of the Flow Duration Curve | [-∞, +∞] | Yilmaz et al. (2008) | Measuring the model's bias in predicting moderate flows (typically fall within the 25th to 75th percentiles). |
| P−factor | Percent of simulation that are without the given uncertainty bounds | [0, 1] | Abbaspour et al. (2007) | Measuring the percentage of reference data falling outside the given uncertainty range |
| **Error metrics (The smaller is better, ideal value is 0)** | | | | |
| RMSE | Root Mean Square Error | [0, +∞] | Kenney & Keeping (1962) | - |
| MSE | Mean Square Error | [0, +∞] | Makridakis et al. (1982) | - |
| ubRMSE | Unbiased Root Mean Square Error | [0, +∞] | Entekhabi et al. (2010) | Remove systematic bias from RMSE and only considers random errors. |
| CRMSE (NRMSE) | Centered Root Mean Square Error (Normalized Root Mean Square Error) | [0, +∞] | Stephen & Kazemi (2014) | Measuring the random component of model error , independent of their mean values. |
| MAE | Mean Absolute Error | [0, +∞] | Yapo et al. (1996) | less sensitive to outliers |
| RSS | Residual sum of squares | [0, +∞] | Archdeacon (1994) | - |
| RSR | RMSE-observations Standard Deviation Ratio | [0, +∞] | Legates & McCabe Jr (1999) | - |
| IPE | The Ideal Point error | [0,1] | Dawson et al. (2012) | - |
| **Correlation metrics (The larger is better. The ideal value is 1)** | | | | |
| R | Correlation Coefficient | [-1, 1] | Pearson (1920) | - |



| R2 | Coefficient of Determination | [0, 1] | Box (1966) and Barrett (1974) | - |
|---|---|---|---|---|
| ubR | Unbiased Correlation Coefficient | [-1 1] | Olkin & Pratt (1958) | Not affected by systematic bias |
| ubR2 | Unbiased Coefficient of Determination | [-1 1] | Olkin & Pratt (1958) | Not affected by systematic bias |
| rSpearman | Spearman's Rank Correlation Coefficient | [-1 1] | Spearman (1961) | Measuring the monotonic relationship between two variables |
| br2 | R-squared multiplied by regression slope | [0, 1] | Krause et al. (2005) and Krstic et al. (2016) | Combines the model's bias and goodness of fit |
| **Efficiency metrics (The larger is better. The ideal value is 1)** | | | | |
| NSE | Nash-Sutcliffe Efficiency | [-∞, 1] | Nash & Sutcliffe (1970) | - |
| LNSE | Log Nash-Sutcliffe Efficiency | [0, 1] | Lamontagne et al. (2020) | More sensitive to lower values |
| mNSE | Modified Nash-Sutcliffe Efficiency | [0, 1] | Legates & McCabe Jr, (1999) | Using absolute differences instead of squared differences |
| rNSE | Relative Nash-Sutcliffe Efficiency | [-∞, 1] | Legates & McCabe Jr, (1999) | Suitable for evaluating relative errors |
| wsNSE | Weighted Seasonal Nash-Sutcliffe Efficiency | [-∞, 1] | Zambrano-Bigiarini & Bellin (2012) | Allows for evaluating model performance across different seasons while considering the relative importance of seasons |
| KGE | Kling-Gupta Efficiency | [-∞, 1] | Gupta et al. (2009) | - |
| KGESS | Standardized Kling-Gupta Efficiency | [-∞, 1] | Knoben et al. (2019) | A normalized version of KGE, facilitating comparison between different models |




| | | | | |
|---|---|---|---|---|
| ubKGE | Unbiased Kling-Gupta Efficiency | [-∞, 1] | Gupta et al. (2009) | Removing bias calculation |
| KGEkm | Kling-Gupta Efficiency with Known Moments | [-∞, 1] | Pizarro & Jorquera (2024) | Considering coefficient of Variation |
| KGElf | Kling-Gupta Efficiency for Low Flows | [-∞, 1] | Garcia et al. (2017) | Evaluating the model's ability to predict low flows |
| **Agreement metric (The larger is better. The ideal value is 1)** | | | | |
| IA | Index of Agreement | [0, 1] | Krause et al. (2005) | - |
| RIA | Relative Index of Agreement | [0, 1] | Krause et al. (2005) | - |
| ReIA | The Refined Index of Agreement | [0, 1] | Willmott et al. (2012) | - |
| valindex | Valid Index | [0, 1] | Criss & Winston (2008) | Measuring the proportion of model predictions falling within an acceptable range |
| L | Likelihood Estimation | [0, 1] | Myung (2003) | Evaluating the probability of model predictions |
| **Others** | | | | |
| rSD | Ratio of Standard Deviations | [-∞, +∞] | Everitt & Skrondal (2010) | Greater than 1 indicates that the simulation has larger variability, vice versa |
| RV | Relative Variability | [-∞, +∞] | Everitt & Skrondal (2010) | - |
| CV | Coefficient of Variation | [-∞, +∞] | Lovie (2005) | - |

Our benchmarking system uses a variety of metrics to evaluate LSM performance thoroughly (**Table 2**). This approach offers different viewpoints on model behavior, detailed comprehension of model strengths and weaknesses, versatile comparison abilities for both individual and inter-model assessments, and efficient implementation using Xarray and Dask software for handling large datasets. The system incorporates various categories of metrics to capture different aspects of model performance. For example, Bias metrics, such as Percent Bias (PB) and Absolute Percent Bias (APB), measure systematic

over- or under-estimation and bias magnitude, respectively. Error metrics, including Root Mean Squared Error (RMSE), Unbiased Root Mean Squared Error (ubRMSE), Centered Root Mean Square Difference (CRMSD), and Mean Absolute Error (MAE) provide different perspectives on the magnitude and nature of model errors. Efficiency metrics like Nash-Sutcliffe Efficiency (NSE) and Kling-Gupta Efficiency (KGE) evaluate model performance relative to baselines and combine multiple



aspects of the model-data agreement. Correlation metrics, including Pearson correlation coefficient and R², quantify the
strength and direction of linear relationships between model outputs and observations. The Index of Agreement provides a
more comprehensive assessment of magnitude and phase agreement. For categorical data, the Kappa coefficient evaluates
agreement while accounting for chance. Variability metrics such as Relative Variability and Percent Change in maximum and
minimum values help identify whether models accurately capture the range of system variability. Bias-corrected versions of
several metrics focus on assessing agreement in variability patterns after removing mean biases. In summary, this
comprehensive approach provides a robust foundation for quantitative LSM assessment, enabling a multi-faceted evaluation
that captures various aspects of the model-observation agreement. By implementing this range of metrics, our benchmarking
system offers a thorough and nuanced evaluation of LSMs, supporting scientific understanding and practical model
improvement.

## 3.2 The scoring index

OpenBench implements a suite of normalized score indices developed in ILAMB (Collier et al., 2018; Arora et al., 2023),
ranging from 0 to 1, with 1 indicating perfect agreement between the model and observations. ILAMB encompasses several
key indices, each designed to evaluate specific aspects of model performance. The Normalized Bias Score (nBiasScore)
quantifies systematic errors in the model's predictions, normalized by observational variability. For a given variable $v(t, x)$,
where $t$ represents time and $x$ represents spatial coordinates. We first calculate the bias from the temporal means of both the





reference $\overline{v_{ref}}(\mathbf{x})$ and model $\overline{v_{sim}}(\mathbf{x})$ data. To score the bias, we normalize it by the centralized root mean square (CRMS) of the reference data:

$$\mathbf{CRMS}(x) = \sqrt{\frac{\int_{t_0}^{t_f}\left(v_{ref}(t,x)-\overline{v_{ref}}(x)\right)^2 dt}{t_f-t_0}}$$

(1)

Where t0 and tf is the first and final timestep, respectively. The relative error in bias is then given as $\boldsymbol{\varepsilon}_{\text{bias}}(x) = \frac{|bias(x)|}{\mathbf{CRMS}(x)}$. The bias score as a function of space is then computed as:

$$\mathbf{nBiasScore}(x) = e^{-\varepsilon_{\text{bias}}(x)}$$

(2)

This score effectively penalizes large biases relative to the natural variability of the system. To evaluate the model's ability to capture observational variability, we employ the Normalized RMSE Score (nRMSEScore). Similar to **nBiasScore**, We first calculate the centralized RSME:

$$\text{CRESM}(x) = \sqrt{\frac{\int_{t_0}^{t_f}\left((v_{sim}(t,x)-\overline{v_{sim}}(x))-\left(v_{ref}(t,x)-\overline{v_{ref}}(x)\right)\right)^2 dt}{t_f-t_0}}$$

(3)

The relative error in bias is then given as $\boldsymbol{\varepsilon}_{\text{rmse}}(x) = \frac{\mathbf{CREMS}(x)}{\mathbf{CRMS}(x)}$. The nRMSEScore as a function of space is then computed as:

$$\mathbf{nRMSEScore}(x) = e^{-\varepsilon_{\text{CRESM}}(x)}$$

(4)

This metric is particularly sensitive to differences in variability patterns between model outputs and observations. For variables with strong seasonal patterns, the Normalized Phase Score (nPhaseScore) assesses the model's ability to capture the timing of seasonal cycles, providing insight into the model's representation of temporal dynamics. The nPhaseScore is calculated as:

$$\mathbf{nPhaseScore}(x) = \frac{1}{2}\left[1 + \cos\left(\frac{2\pi\theta(x,\lambda,\phi)}{nstep}\right)\right]$$

(5),

where $\boldsymbol{\theta}(x, \lambda, \phi)$ is the time difference between modeled and observed maxima:



$$\theta(x, \lambda, \phi) = \text{maxima}\big(c_{sim}(x, t, \lambda, \phi)\big) - \text{maxima}\big(c_{ref}(x, t, \lambda, \phi)\big)$$

(6).

Here $c_{sim}$ and $c_{ref}$ are the climatological mean cycles of the model and reference data, maxima( ) calculates the maximum value at the grid $x$ of evaluation time resolution. The division by nstep (e.g., 12 for monthly or 365 for daily resolution) normalizes the phase difference to the annual cycle. Interannual variability, a critical aspect of climate modeling, is evaluated using the Normalized Interannual Variability Score (nIavScore). nIavScore is given by first removing the annual cycle from both the reference and model:

$$iav_{ii}(x) = \sqrt{\frac{\int_{t_0}^{t_f}\big(v_{ii}(t,x) - c_{ii}(t,x)\big)^2 dt}{t_f - t_0}}$$

(7),

where ii represents sim or ref. Then, the relative error is calculated as

$$\varepsilon_{iav}(x) = \frac{iav_{\text{sim}}(x) - iav_{\text{ref}}(x)}{iav_{\text{ref}}(x)}$$

(8).

Similar to Equation (2) and (4), the nIavScore is given by

$$\textbf{nIavScore}(x) = e^{-\varepsilon_{\text{iav}}(x)}$$

(9).

This score is crucial for assessing the model's performance in representing year-to-year variations driven by climate factors. The Spatial Score (nSpatialScore) is used to evaluate how well the model captures the spatial distribution of a variable compared to observations. To provide an overall assessment of model performance, we calculate an Overall Score (OvScore) that combines these individual metrics. This composite score gives double weight to the nRMSEScore due to its importance in capturing both bias and variability aspects, which is consistent with ILAMB. The Relative Score (ReScore) is designed to compare performance across simulations by normalizing a model's overall score relative to the multi-simulation mean and standard deviation. Positive values indicate above-average performance, while negative values indicate below-average performance. Detailed information can be obtained from Collier et al. (2018).

ILAMB and OpenBench exhibit two key differences in their scoring methodologies. The first distinction lies in their approach to calculating global mean scores. ILAMB applies mass weighting when evaluating variables that represent carbon or water mass/flux, such as Gross Primary Production (GPP) or precipitation. This method can lead to global mean scores being





disproportionately influenced by middle and low latitudes, as exemplified by the significant impact of GPP or precipitation in the Amazon. In contrast, OpenBench offering greater flexibility in its weighting methods. OpenBench supports multiple weighting options that users can select based on their requirements. Users can choose between a simple spatial integral for unweighted averaging, area weighting to account for varying grid cell sizes across latitudes, or mass weighting for mass/flux variables. This flexibility allows researchers to choose the best weighting method for their particular analysis. For example, when evaluating GPP, researchers might opt for mass weighting to align with ILAMB's methodology, or they could choose area weighting to ensure more balanced representation across latitudes. The choice of weighting method can significantly impact the final results, particularly when analyzing variables with strong spatial heterogeneity. The second major difference pertains to how these systems handle multiple reference datasets. ILAMB combines evaluation results from different reference datasets, assigning weights to each and combining them multiplicatively to produce a single final score that incorporates all datasets. OpenBench, on the other hand, provides users with multiple reference datasets and allows them to select one or more that they consider most accurate. It then reports scores separately for each chosen dataset without applying weights. This approach gives users more flexibility and transparency in interpreting results, allowing them to make informed decisions based on their knowledge of dataset quality and relevance to their specific research questions. These methodological differences reflect the distinct philosophies and goals of each system. ILAMB's approach emphasizes a comprehensive, weighted assessment that accounts for the relative importance of different regions and datasets. OpenBench prioritizes user choice and equal spatial representation in its scoring methodology, allowing for a more customizable and potentially more equitable evaluation process. Both approaches have their merits, and the choice between them may depend on the specific needs and preferences of the research community using these benchmarking tools.

In summary, by combining multiple normalized scores that assess different aspects of model performance, we enable a nuanced understanding of model strengths and weaknesses. This approach not only supports the evaluation of individual models but also facilitates inter-model comparisons and the tracking of model improvements over time.

**3.3 Datasets**

OpenBench integrates a diverse array of benchmarking data spanning multiple variables, levels, and spatiotemporal resolutions. This approach ensures a thorough evaluation of modern high-resolution LSMs, which require increasingly detailed and accurate input data to capture complex land-atmosphere interactions.

The strength of OpenBench lies in its extensive collection of baseline datasets, categorized into five main groups: radiation and energy cycle, ecosystem and carbon cycle, hydrology cycle, parameters and atmospheric forcing, and human activity. These datasets are derived from five primary sources: field observations, satellite remote sensing, reanalysis data, machine learning, and model outputs. Each source offers unique advantages, contributing to a more comprehensive understanding of land surface processes. Field observations provide high-accuracy, ground-truth data essential for model validation and calibration. While often limited in spatial coverage, these datasets offer unparalleled accuracy and temporal resolution. Satellite





remote sensing delivers extensive spatial coverage and consistent temporal sampling, which are crucial for monitoring large-scale land surface processes. Reanalysis data combines model simulations with observations to create consistent, gridded data products, particularly useful for long-term study or filling observational gaps. Model outputs and machine learning, while not direct observations, provide estimates of variables that are challenging to measure directly.

The spatial-temporal scope of OpenBench's datasets is another critical feature. Many datasets span several decades, allowing
for evaluating long-term trends and interannual variability. This extended temporal coverage enables assessing LSMs performance over long historical periods. The resolution ranges from coarse (e.g., 0.5° for ILAMB datasets (Collier et al., 2018)) to very fine (e.g., 500m for MODIS-based products (Varquez et al., 2021)), making it possible to evaluate. This range allows for evaluating LSMs across different spatial scales, from global assessments to regional or plot scale studies.

A unique aspect of OpenBench is its inclusion of datasets focused on human impacts on land surface processes. This approach
recognizes the growing importance of anthropogenic factors in shaping the Earth system. Datasets like AH4GUC (Varquez et al., 2021), which provides global anthropogenic heat flux data, and GDHY (Iizumi and Sakai, 2020), offering detailed information on global crop yields, enable the evaluation of urban heat island effects and agricultural impacts in LSMs.

OpenBench's inclusion of multiple datasets for each variable allows for a more robust evaluation of LSMs. This multi-dataset approach enables users to assess model performance against a range of reference data, providing a more comprehensive
evaluation. For instance, in evaluating evapotranspiration, OpenBench includes datasets like GLEAM4.1 (Miralles et al., 2011), FLUXCOM (Jung et al., 2019), X-BASE (Nelson et al., 2024), Xu2024 (Xu et al., 2024), and ERA5-Land (Muñoz-Sabater et al., 2021), each with its own methodology and characteristics. Users can assess model performance across multiple variables simultaneously, identifying potential compensating errors or cross-variable inconsistencies that might be missed when evaluating single variables in isolation. This multi-dimensional approach provides a more complete picture of model
performance and helps guide future model development efforts. Meanwhile, OpenBench's dataset collection is designed to be expandable and updateable, ensuring its relevance in the rapidly evolving field of Earth system science. As new datasets become available or existing datasets are updated, they can be seamlessly integrated into the OpenBench framework.

It is noted that while OpenBench integrates a comprehensive collection of datasets, we cannot directly provide specific data due to copyright restrictions and licensing agreements. However, to ensure transparency and reproducibility, we have included
relevant links to the original data sources in **Table S1-S5**. These links will guide users to the appropriate platforms to access the datasets following the respective terms and conditions set by the data providers. To demonstrate the functionality and structure of OpenBench, we have included a set of self-generated sample data. This sample data mimics the characteristics and format of the actual datasets, allowing users to familiarize themselves with the OpenBench framework and its capabilities without infringing on any copyright issues. We encourage users to utilize these sample datasets for initial testing and
exploration of the OpenBench system and then proceed to acquire the complete datasets from the original sources for comprehensive model evaluations.



### 3.4 Supporting models

OpenBench has been designed to accommodate a diverse array of land surface models, facilitating comprehensive
intercomparison and evaluation studies. This multi-model support is a key feature of OpenBench, enabling researchers to
assess and compare the performance of various models across different land surface processes and variables. Currently,
OpenBench supports various state-of-the-art land surface models, including multiple versions of the Common Land Model
(CoLM2014 and CoLM2024) (Bai et al., 2024; Dai et al., 2003; Fan et al., 2024), the Community Land Model Version 5
(CLM5) (Lawrence et al., 2019), Noah-MP 5.0 (He et al., 2023), the Minimal Advanced Treatments of Surface Interaction and
Runoff model (Version 2021) (Nitta et al., 2014), Atmosphere-Vegetation Interaction Model (AVIM) (Li et al., 2002), the
Global Land Data Assimilation System (GLDAS2) (Rodell et al., 2004), Today's Earth (TE)
(https://www.eorc.jaxa.jp/water/index.html) and the Variable Infiltration Capacity (VIC) Model (Hamman et al., 2018) and so
on. OpenBench has expanded its capabilities to support Lambert Conformal projections outputs from regional climate models
such as the Climate extension of the Weather Research and Forecasting model (CWRF) (Liang et al., 2012) and the Weather
Research and Forecasting (WRF) model (Lo et al., 2008).

Furthermore, OpenBench supports CMIP-styled series outputs, such as the Coupled Model Intercomparison Project (CMIP)
style simulation, such as LS3MIP (Van Den Hurk et al., 2016) and ISIMIP (Wartenburger et al., 2018), allowing for seamless
integration of global climate model data into the evaluation framework. Each supported model is integrated into the system
through a dedicated namelist file that maps the model's output variables to standardized variables used within OpenBench.
This approach ensures consistent comparison and evaluation across different models, regardless of their native output format
or projection.

### 3.5 Case studies

To illustrate the analytical capabilities of our evaluation system, we present comprehensive case studies focusing on two critical
aspects of hydrological modeling: river discharge evaluation and inundation fraction assessment. These analyses were
conducted using simulations from the CaMa-Flood Version 4.22 model (Yamazaki et al., 2013), driven by 0.25° remapped
daily runoff data from the Global Reach-level Flood Reanalysis (Yang et al., 2021). The evaluation was performed globally
with 0.25° spatial resolution, utilizing observational data from the Global Runoff Data Centre (GRDC) for discharge validation
and the GIEMS dataset (Prigent et al., 2020) for inundation fraction assessment. Our analysis demonstrates the system's
versatility in conducting site-specific and global-scale evaluations. **Figure 2a** presents a detailed comparison of simulated and
observed streamflow hydrographs for a representative station. This station-level analysis reveals the model's strong
performance in capturing both the magnitude and temporal variability of streamflow patterns. The close alignment between
simulated and observed values indicates robust model performance at the local scale. **Figure 2b** illustrates the spatial
distribution of KGESS values for simulated discharge across the globe. The analysis reveals distinct regional patterns in model
performance. The model demonstrates particularly strong capabilities in simulating discharge across wet regions, including



the Amazon basin, Japan, and the Eastern United States. However, performance metrics indicate lower accuracy in the Western United States, Central Australia, and Southern Africa. These regional variations can be attributed to several factors, including the influence of human activities, uncertainties in precipitation datasets, and limitations in model parameterization schemes (Wei et al., 2020). The impact of human activities on model performance is particularly evident in regions like the Western United States, where streamflow patterns are significantly modified by dam operations (Hanazaki et al., 2022). This finding

underscores the importance of incorporating human water management practices in regions with intensive anthropogenic influence to achieve reliable simulation results. **Figure 2c** presents global patterns of correlation coefficients for simulated inundation fraction. The results indicate strong model performance in low-latitude regions, particularly in the Amazon basin and South Asia. However, significant discrepancies emerge in high-latitude areas (above 60°N). This spatial pattern of model performance highlights the need for improved representation of snow-related processes and precipitation phase partitioning in

these regions (Jennings et al., 2018).

OpenBench implements automated grouping of metrics and scores according to both IGBP and PFT classifications to provide a comprehensive evaluation of model performance across diverse ecological zones. **Figure 3** presents a detailed heatmap visualization of performance indices categorized by IGBP land cover types, based on CoLM2024 simulations evaluated against X-BASE reference data (Nelson et al., 2024) for 2002-2003. The analysis incorporates six fundamental performance scores

developed within the ILAMB framework, as detailed in **Sect. 3.2**. The visualization reveals several significant patterns in model performance across different ecosystems. The overall nPhaseScore of 0.69 demonstrates the model's robust capability in capturing seasonal variations across all biomes. Particularly noteworthy is the model's exceptional performance in forest ecosystems, where Evergreen Needleleaf Forests (ENF), Deciduous Needleleaf Forests (DNF), and Mixed Forests (MF) exhibit consistently high nPhaseScores. These results indicate the model's sophisticated ability to simulate the complex

dynamics of multi-layered forest ecosystems. However, the analysis also identifies specific challenges in certain environmental contexts. The model's performance notably decreases in extreme environments, with lower scores across multiple metrics for Snow and Ice (SNO) and Barren or Sparsely Vegetated (BSV) regions. Additionally, Evergreen Broadleaf Forests (EBF) show particularly low nBiasScores, reflecting substantial magnitude discrepancies between simulated and observed values. This finding underscores the persistent challenges in accurately modeling these data-sparse, highly dynamic ecosystems. These

insights have important implications for model application across different research contexts. Researchers focusing on temperate and boreal forest ecosystems can proceed with high confidence in the model's capabilities. However, studies targeting arid regions, snow-covered areas, or tropical rainforests should incorporate additional validation steps and exercise greater caution in interpreting results. This systematic evaluation across biomes thus provides essential guidance for appropriate model application in diverse ecological settings.



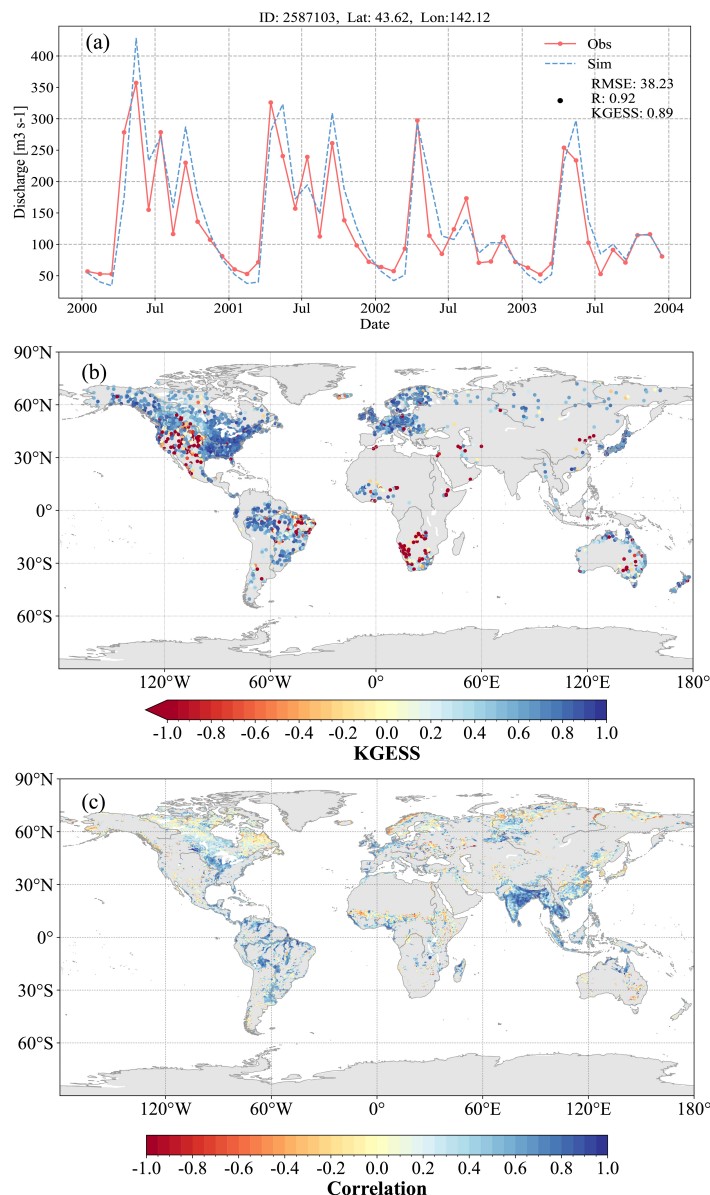

**Figure 2: Example of river discharge evaluation: (a) simulated and observed streamflow hydrographs for an example station; (b) global maps of KGESS values simulated discharge dataset; and (c) global maps of R values simulated inundation dataset.**



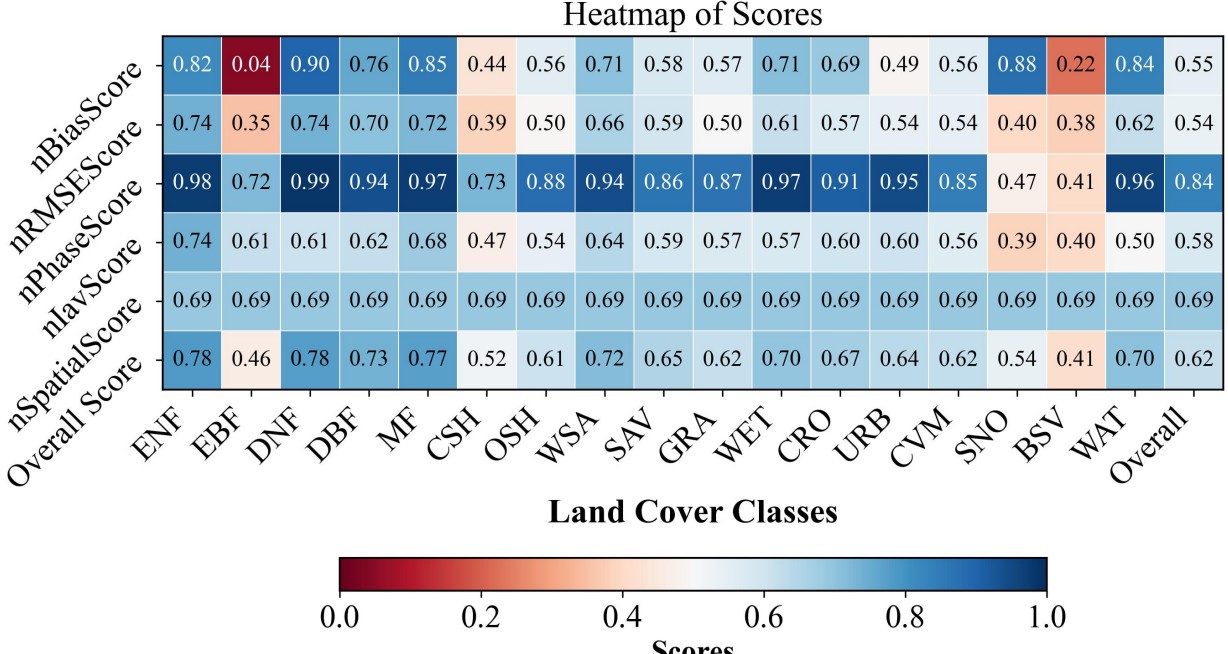

Figure 3: An example of heatmap of score indexes for GPP classified by IGBP land cover

**Figure 4** demonstrates OpenBench's capability to evaluate anthropogenic influences on urban thermal environments through a detailed comparison of CoLM2024 simulations with AH4GUC observational data for Southeast Asia. The analysis reveals generally strong agreement between simulated and observed anthropogenic heat flux patterns across most regions. However, notable discrepancies emerge in specific areas, particularly the corridor extending from central China to northern Vietnam and regions near Laos, where negative correlations indicate potential systematic biases in model representation. While the precise mechanisms driving these regional differences remain under investigation, these findings highlight the importance of refined urban parameterization schemes in land surface models. The system's evaluation capabilities extend beyond thermal processes to encompass multiple aspects of human-environment interactions. Through a comprehensive assessment of variables, including latent heat, albedo, and surface temperature changes, OpenBench provides valuable insights into the complex relationships between anthropogenic activities and land surface processes, guiding improvements in their model representation.

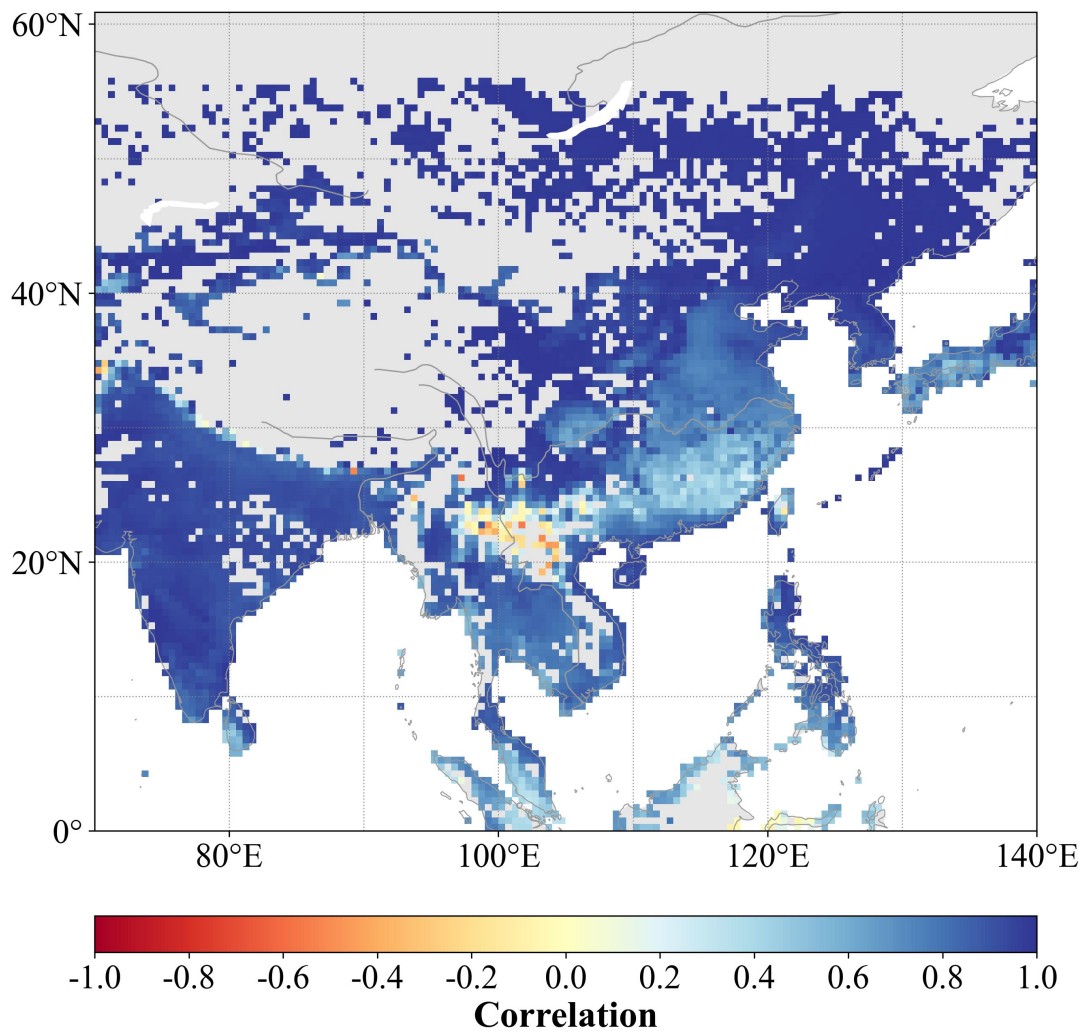

**Figure 4: Pearson's correlation coefficient between CoLM2024 simulation and AH4GUC generation for urban anthropogenic heat flux over Southeast Asia.**

**Figure 5** presents a detailed analysis of agricultural modeling capabilities, comparing CoLM2024 simulated corn yields with GDHY-generated observational data across the United States. The analysis reveals distinct regional patterns in model performance: approximately 20% yield underestimation in the western United States, significant overestimation in central regions, and notable underestimation in eastern areas. These spatial patterns of bias may stem from multiple sources, including uncertainties in the GDHY observational dataset and the CoLM2024 model structure. Particularly noteworthy are the substantial differences in planted area and crop distribution between the two datasets, indicating fundamental challenges in representing agricultural systems within current modeling frameworks. These findings underscore the significant opportunities for advancement in both modeling and observational approaches to crop yield estimation. Future research efforts should focus





on reducing uncertainties in simulation and observational datasets while improving the representation of agricultural processes in land surface models.

In summary, these case studies demonstrate the comprehensive analytical capabilities of our evaluation system. Through its ability to conduct detailed analyses across multiple spatial scales and variables, OpenBench provides researchers with powerful tools for assessing model performance and identifying specific areas for improvement. This multi-scale, multi-variable approach supports theoretical understanding and practical application of land surface models, ultimately contributing to enhanced representation of Earth system processes.

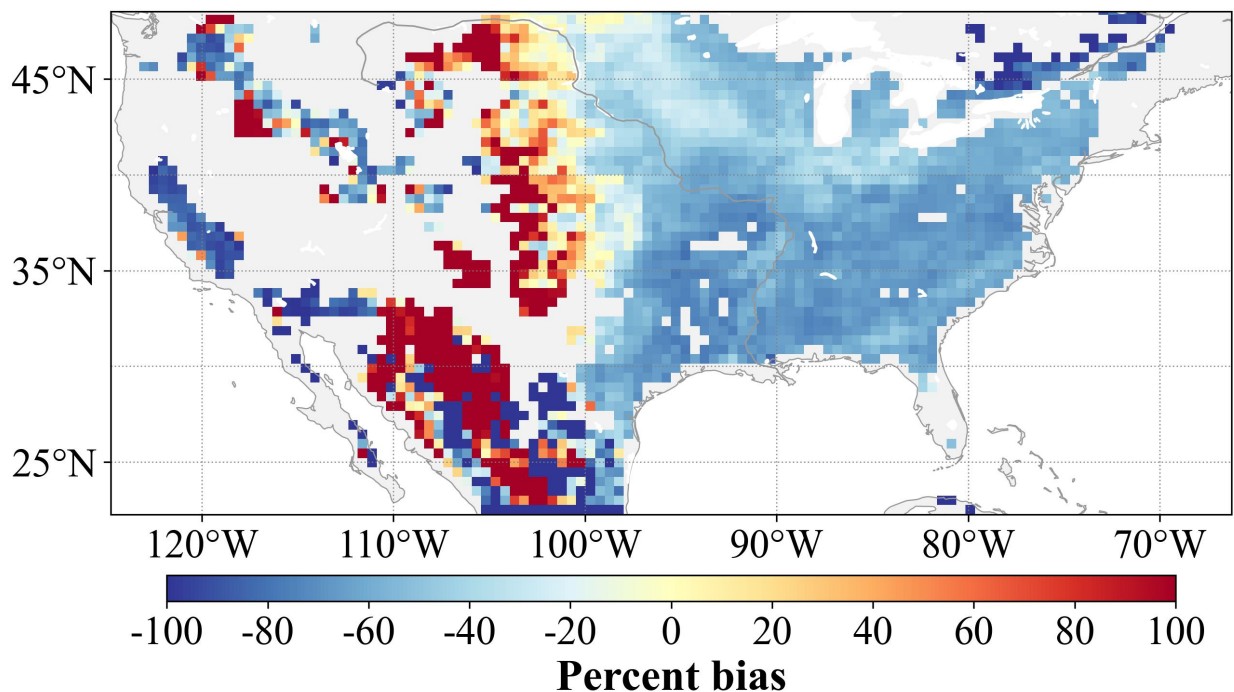


**Figure 5: Percentage bias between CoLM2024 simulated and GDHY generated crop yield of corn for the United States.**

## 4 Comparisons

### 4.1 Overview

OpenBench offers a comprehensive suite of comparison capabilities designed to facilitate a thorough evaluation of model performance across diverse scenarios, land cover types, and temporal scales. The system incorporates several key functionalities that enable sophisticated analysis while maintaining user accessibility and scientific rigor.

The framework's evaluation architecture encompasses multiple complementary approaches to model assessment. At its foundation, ecosystem-based comparisons allow researchers to evaluate performance across different IGBP and PFT land



cover classifications, providing crucial insights into model behavior within specific ecological contexts. This capability is enhanced by multi-metric visualization tools, including heat maps, Taylor diagrams, and target diagrams, which offer intuitive yet comprehensive overviews of model capabilities by simultaneously displaying multiple statistical metrics for model-observation comparisons. To support detailed analysis of model behavior, OpenBench implements advanced distribution and pattern analysis tools. These include kernel density estimation plots and parallel coordinate plots, which facilitate the

comparison of metric distributions across models and enable identification of patterns in multivariate performance data. The system's temporal performance evaluation capabilities, implemented through seasonal portrait plots, provide detailed insights into variations in model accuracy across different seasonal cycles. Statistical analysis within OpenBench is supported by robust summary tools, including box and whisker plots that offer concise yet comprehensive overviews of model performance across different metrics and scenarios. This statistical framework ensures that comparisons remain objective and scientifically sound

while presenting results in an accessible format for interpretation.

The implementation of multiple model comparisons follows a systematic and efficient approach. The process begins with standardization of model outputs through a sophisticated data processing pipeline, capable of handling various input formats and temporal/spatial resolutions. The comparison processing module orchestrates this analysis through support for multiple comparison methods, with parallel processing capabilities implemented via the Joblib library to ensure computational

efficiency. Evaluation items and reference sources systematically organize results from the comparison process within a structured output directory. The system automatically generates comparison artifacts, including metrics and score files, which form the basis for comprehensive visualization and analysis. This structured approach ensures that adding new models to the comparison framework requires minimal effort, typically involving only the update of simulation namelists with new model information and data sources. This integrated approach to model comparison and evaluation provides researchers with

powerful tools for understanding model behavior while maintaining the flexibility needed to address diverse research questions in land surface science. The system's design philosophy emphasizes scientific rigor and practical utility, ensuring that comparative analyses can be conducted efficiently while maintaining the highest standards of scientific validity.

## 4.2 Case studies

To demonstrate the comprehensive capability of our evaluation system, we present several case studies to demonstrate the

ability of the evaluation system to compare between modes, compare between different parameterized schemes, and compare between CMIP-styled datasets. It's important to note that our primary goal is to showcase the evaluation system's functionality rather than to make definitive judgments about any particular model's performance. These case studies are practical examples of the system's versatility and analytical power.





### 4.2.1 Multiple models comparison

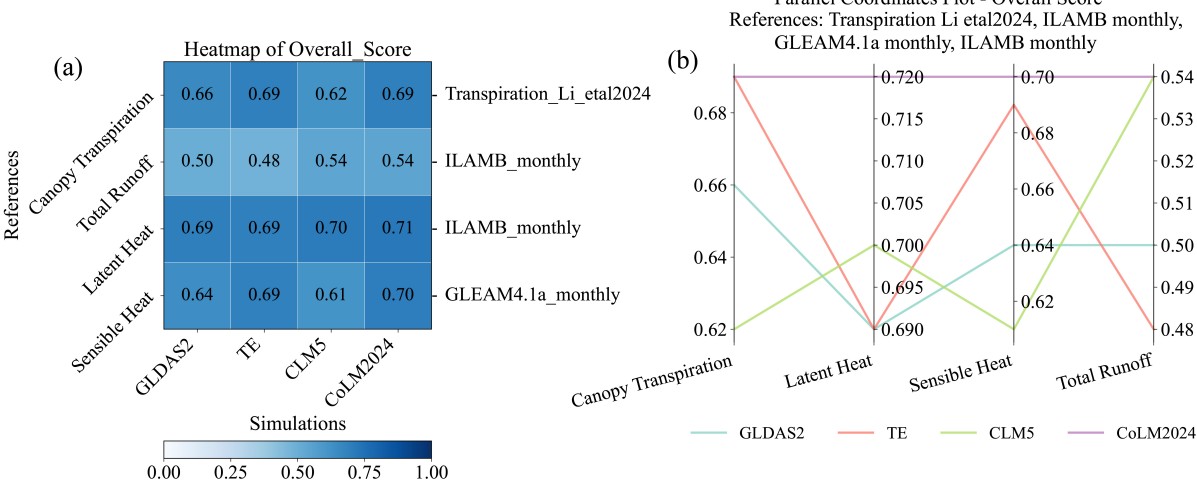


**Figure 6: Overall score comparisons of sensible heat, latent heat, total runoff, canopy transpiration using (a) heat map and (b) parallel coordinates approaches for GLDAS2, CLM5, TE, and CoLM2024.**




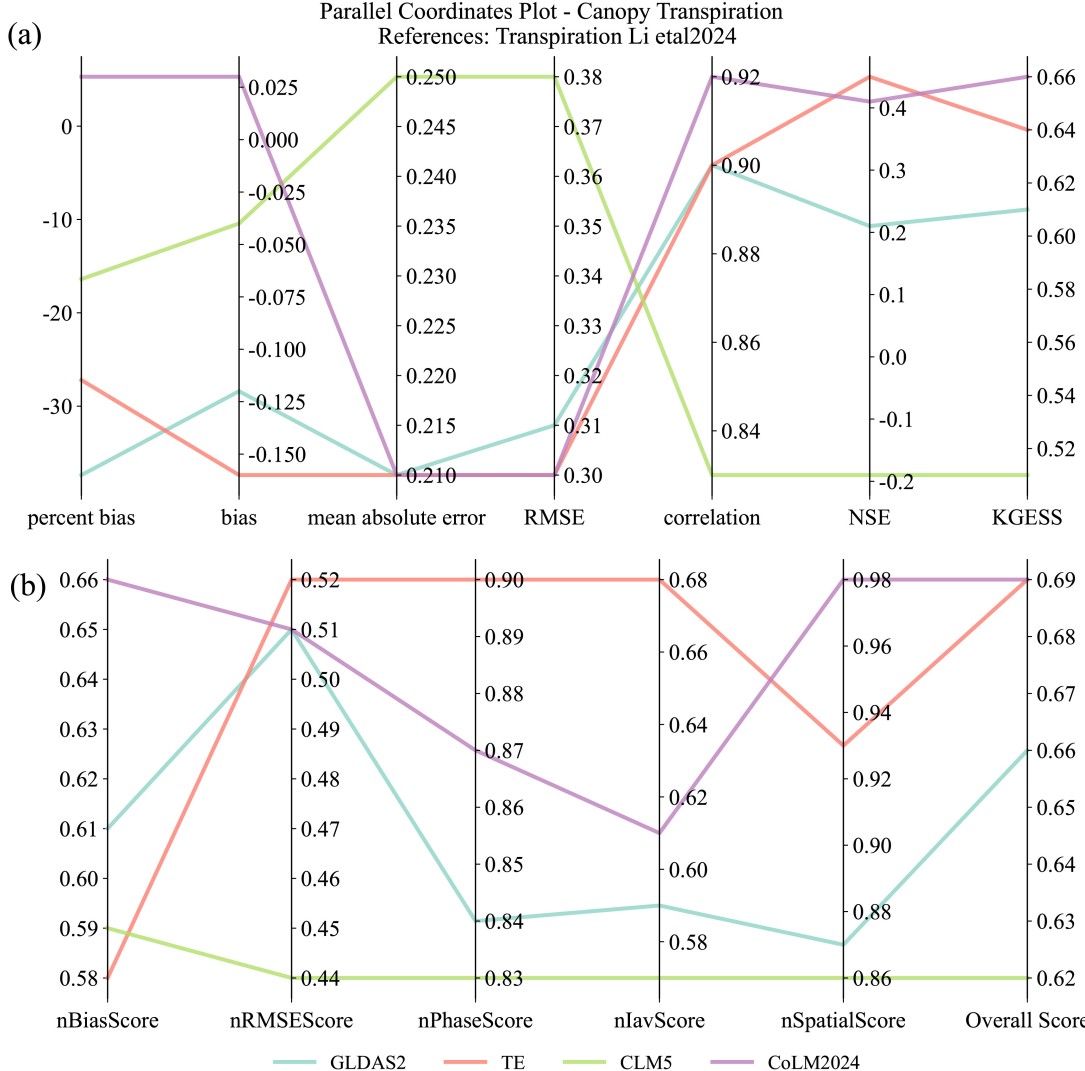

**Figure 7: Evaluation of canopy transpiration using various (a) metric and (b) score indexes for GLDAS2, CLM5, TE, and CoLM2024.**

To demonstrate the multiple models' analytical capabilities of OpenBench, we conducted a comparative analysis of four state-of-the-art land surface models: GLDAS2, CLM5, TE, and CoLM2024. The evaluation period spanned from 2002 to 2006, utilizing monthly temporal resolution. Multiple reference datasets were incorporated, including Li et al. (2024) for canopy transpiration, the FLUXCOM dataset from ILAMB, GLEAM4.1a for surface heat fluxes, and LORA from ILAMB for total runoff assessment.

**Figure 6** illustrates the comparative analysis through two complementary visualization approaches: a heat map and a parallel coordinates plot. The heat map (left panel) provides an intuitive visualization of relative model performance across different




variables, while the parallel coordinates plot (right panel) reveals intricate relationships between various performance metrics.
This dual visualization strategy enables researchers to quickly identify patterns and trade-offs in model performance across multiple variables simultaneously. The analysis reveals that under current configurations, CoLM2024 and TE demonstrate superior performance in simulating canopy transpiration and total runoff, while CoLM2024 maintains a slight advantage in other variables.

For detailed variable-specific analysis, **Figure 7** presents an in-depth examination of canopy transpiration across all models,
utilizing both conventional metrics (**Fig. 7a**) and normalized scores (**Fig. 7b**). The metrics analysis reveals that CoLM2024 exhibits a tendency to overestimate canopy transpiration, while other models show varying degrees of underestimation, as indicated by the percent bias metric. Furthermore, CoLM2024 achieves optimal performance regarding RMSE minimization, correlation maximization, and KGESS optimization. TE demonstrates particularly strong performance in NSE and ranks second in KGESS. Regarding scoring indices, TE excels in nRMSEScore, nPhaseScore, and nIavScore, whereas CoLM2024
achieves the highest nBiasScore overall score.

This comprehensive comparative analysis not only highlights the relative strengths and weaknesses of each model but also offers valuable insights into their simulation capabilities regarding various aspects of land surface processes. Such a detailed evaluation helps identify areas where models excel or need further refinement, effectively guiding future development efforts.

### 4.2.2 Multiple parameterizations and multiple references

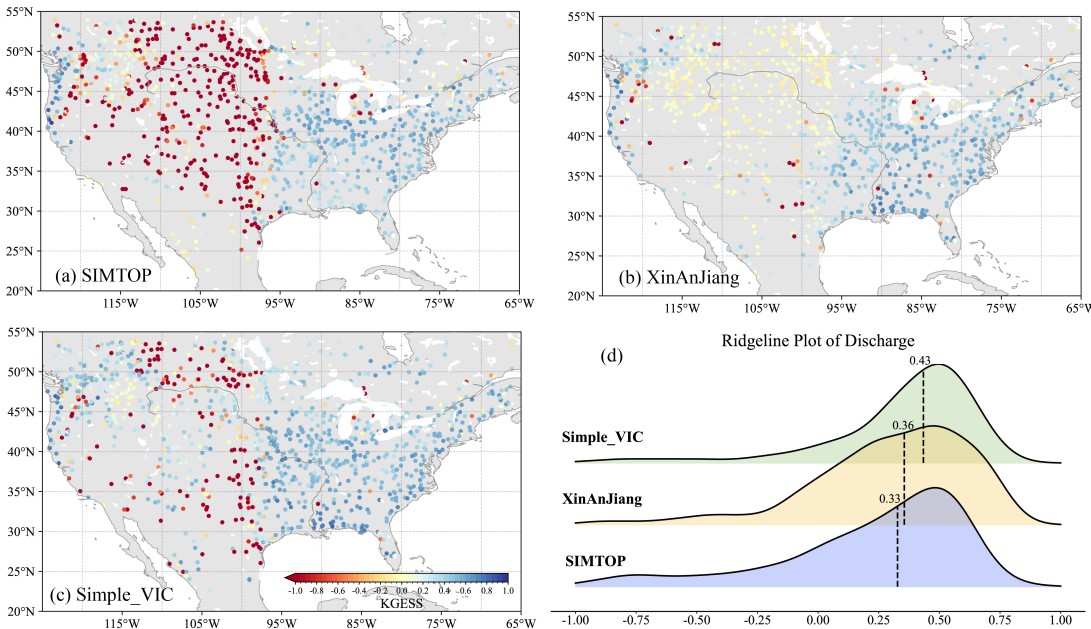


**Figure 8: Comparison of daily discharge simulated from different parameterizations of the CoLM model's runoff generation scheme with GRDC observations using KGESS metric.**



To evaluate the versatility of OpenBench in analyzing model parameterization schemes, we conducted a comprehensive assessment of different runoff generation parameterizations within the CoLM model framework. The analysis focused on daily

discharge simulations at 0.1° resolution from 1985 to 1999, comparing three distinct parameterization approaches: SIMTOP, XinAnjiang, and Simple VIC schemes. These simulations were evaluated against observational data from GRDC.

**Figure 8** presents a spatial analysis of model performance using the KGESS metric across the continental United States. The station-based visualization (**Figs. 8a-c**) reveals distinct spatial patterns in model performance for each parameterization scheme. The Simple VIC parameterization demonstrates superior performance across most regions, particularly in areas with complex

hydrological processes. In contrast, the XinAnJiang scheme exhibits notable strengths in simulating discharge patterns within arid and semi-arid regions, suggesting its particular effectiveness in water-limited environments.

To further elucidate the statistical characteristics of these parameterizations, we employed a ridgeline plot analysis (**Fig. 8d**). This visualization technique effectively captures the distribution of performance metrics across different schemes, with the dashed lines and accompanying numbers indicating median values for each parameterization. The analysis confirms that the

Simple VIC parameterization achieves the highest overall performance metrics, though each scheme shows specific regional strengths.

OpenBench's capability to handle multiple reference datasets is demonstrated through a detailed evaluation of latent heat simulations. **Figure 9** illustrates this multi-reference analysis framework, comparing CoLM simulations against four distinct reference sources: satellite-derived products (CLASS), machine learning outputs (FLUXCOM), in-situ measurements

(PLUMBER2), and reanalysis data (ERA5Land). This comparison was conducted at a monthly temporal resolution and 0.5° spatial resolution for the period 2002-2006. The resulting heat map visualization reveals strong model-data agreement across all reference datasets, with correlation coefficients consistently exceeding 0.90.

This comprehensive evaluation approach validates the model's performance against multiple independent data sources, as well as provides insights into the structural uncertainties inherent in different observational datasets. Such multi-reference validation

is particularly valuable for variables where direct measurements are sparse and each observational approach has its own uncertainties and biases. The consistently high correlation values across different reference datasets enhance confidence in the model's ability to capture fundamental physical processes while also highlighting areas where uncertainties in observational data may impact validation efforts.





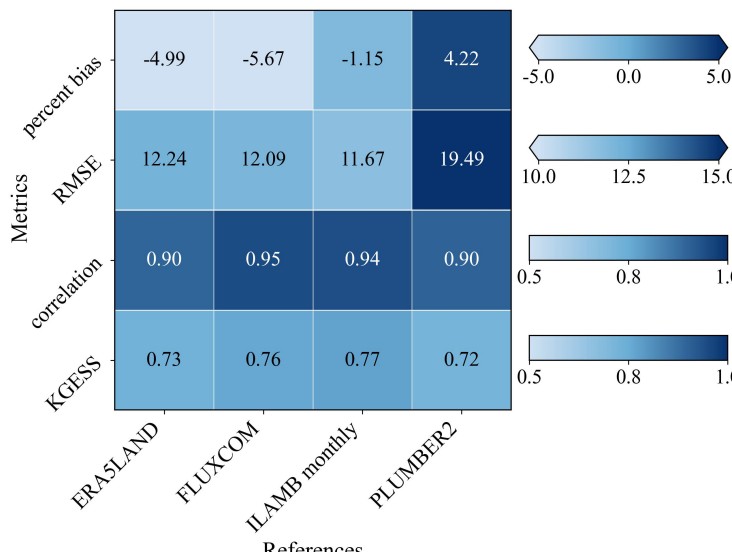

**Figure 9: Evaluation of latent heat flux simulated by CoLM2024 using various metrics with different reference datasets.**

### 4.2.3 CMIP styles comparison

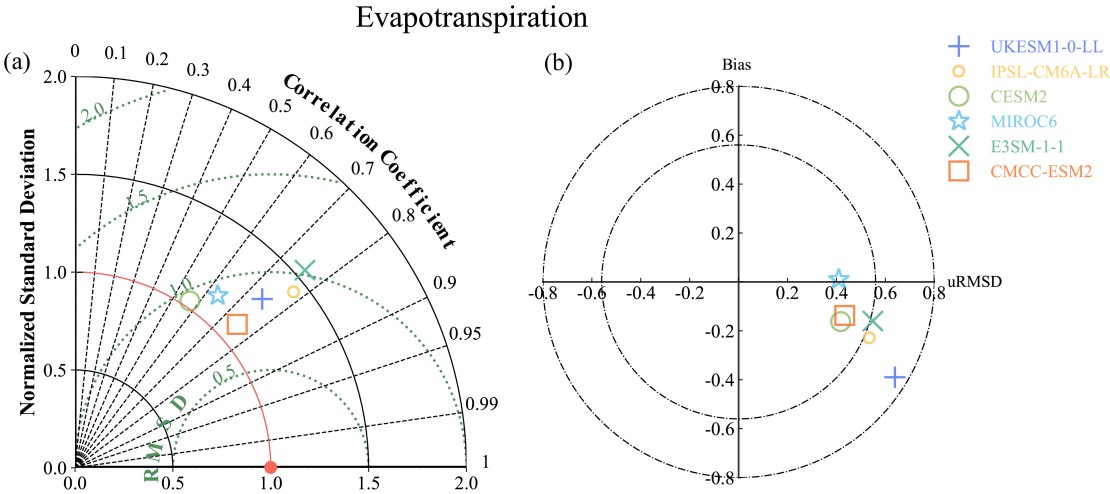

**Figure 10: The Taylor (a) and target diagram (b) for comparing evapotranspiration among the six models in LS3MIP.**



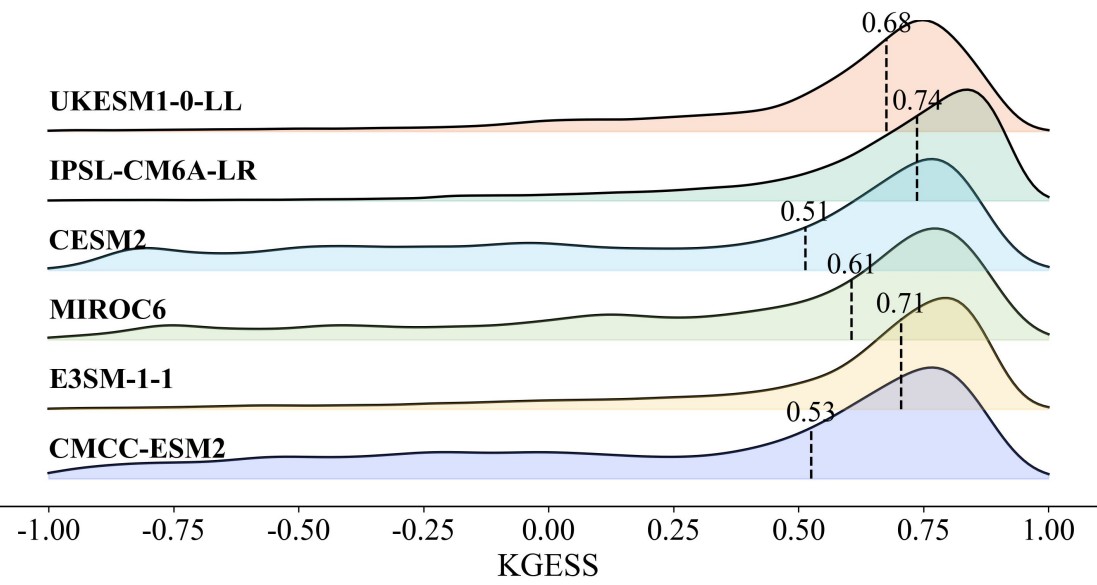

**Figure 11: The ridgeline plot comparing evapotranspiration for the six models in LS3MIP Land-hist experiment.**

OpenBench's evaluation framework incorporates robust capabilities for analyzing CMIP-styled datasets, with particular emphasis on experimental outputs from initiatives such as ISIMIP and LS3MIP. The system's architecture includes specialized

data processing modules designed to handle the standardized conventions of CMIP outputs, including variable naming conventions, temporal frequencies, and grid structures, ensuring seamless integration with the evaluation framework.

**Figure 10** demonstrates this capability through a comprehensive analysis of evapotranspiration simulations from the LS3MIP experiment. The analysis employs both Taylor and target diagrams to provide complementary perspectives on model performance. The Taylor diagram (**Fig. 10a**) effectively visualizes the relationship between correlation coefficients,

normalized standard deviations, and centered root mean square differences. This multi-metric representation enables immediate identification of models that achieve optimal balance across these key performance indicators. The target diagram (**Fig. 10b**) supplements this analysis by providing additional insight into bias components and pattern variations, with distinct symbols differentiating between the various LS3MIP simulations.

To further elucidate the performance distribution across different models, **Figure 11** presents a ridgeline plot analysis of the

KGESS metric. This visualization technique reveals the full spectrum of model performance, highlighting both central tendencies and variations in simulation quality. The analysis demonstrates that while certain models consistently achieve higher performance metrics, considerable variation in simulation quality exists across the ensemble. This variation provides valuable insights into the structural uncertainties inherent in current land surface modeling approaches.

The integration of CMIP-style evaluation capabilities within OpenBench serves multiple critical functions in the broader

context of Earth system modeling. First, it enables systematic assessment of land surface processes within coupled climate





models, providing essential feedback for model development and improvement. Second, it facilitates direct comparisons between offline land surface model simulations and their behavior within coupled frameworks, helping to identify potential interactions and feedback that may affect model performance. Finally, this capability supports comprehensive model intercomparison studies, contributing to our understanding of model uncertainties and their implications for future climate
projections.

This robust framework for evaluating CMIP-style outputs positions OpenBench as a valuable tool for both model development and climate change research. By providing standardized, comprehensive evaluation metrics for these complex datasets, OpenBench supports the ongoing effort to improve our understanding and prediction of land surface processes in the context of global climate change.

## 535    5 Extensibility and Customization

OpenBench is engineered with extensibility and customization as core design principles, enabling the system to evolve alongside the rapidly advancing field of land surface science. This flexible architecture accommodates the integration of new models, variables, datasets, measurement units, evaluation metrics, and scoring systems while maintaining operational consistency and scientific rigor. The system's modular design facilitates seamless incorporation of new reference datasets
through a streamlined configuration process. Researchers can integrate additional observational or reanalysis data by creating appropriate entries in the reference configuration file, specifying dataset locations and characteristics. This process involves defining dataset properties, including directory structures, temporal and spatial resolutions, and variable-specific parameters. For datasets with unique characteristics, users can develop custom processing scripts that integrate smoothly with the existing evaluation framework.

Variable integration follows a similarly structured approach. Adding new variables requires coordinated updates to both reference and simulation configuration files, alongside corresponding dataset configurations that define variable properties. This process may include the development of specialized evaluation metrics and visualization components to effectively represent and analyze the new variables within the system's analytical framework. The integration of new land surface models demonstrates OpenBench's architectural flexibility. Users can incorporate additional models by creating model-specific
namelist files that establish straightforward mappings between model outputs and OpenBench's standardized variables. This integration is supported by updates to the simulation configuration and, where necessary, the development of custom variable filtering scripts to handle model-specific output characteristics. OpenBench's unit conversion system exemplifies its sophisticated approach to extensibility. The unit processing module employs a flexible design that readily accommodates new measurement units for existing and new variables. Users can implement additional unit conversions by creating methods within
the designated class, following established naming conventions. The system's dynamic method calling architecture ensures that new unit conversions integrate seamlessly into the evaluation workflow without requiring modifications to other system components. The system's evaluation framework maintains equal flexibility in incorporating new metrics and scoring





methodologies. Users can implement additional evaluation metrics by creating new methods within the metrics class, properly handling missing data, and maintaining comprehensive documentation. Similarly, new scoring systems can be integrated into the scores class within Mod_Scores.py, with appropriate attention to normalization procedures and interpretation guidelines. This comprehensive approach to extensibility guarantees that OpenBench stays at the forefront of land surface model evaluation capabilities. As new scientific questions emerge, new models are developed, and new observational datasets become available, the system can readily adapt to incorporate these advances. This flexibility is essential for maintaining a state-of-the-art evaluation framework that effectively serves the evolving needs of the land surface modeling community while ensuring consistent, high-quality analysis across various applications and research contexts.

## 6 Conclusions

Our newly developed OpenBench represents a significant advancement in land surface model evaluation methodology, addressing critical gaps in existing evaluation frameworks while introducing innovative capabilities for comprehensive model assessment. By integrating high-resolution benchmark datasets, sophisticated evaluation metrics, and efficient data handling mechanisms, OpenBench provides users with a powerful tool for enhancing the understanding and performance of land surface models. The system's key strengths lie in several areas. First, its ability to handle diverse data types and formats, from station-based measurements to gridded products, enables comprehensive evaluation across multiple spatial and temporal scales. Second, incorporating human activity impacts into the evaluation framework fills a crucial gap in current assessment tools, allowing for a more realistic evaluation of model performance in anthropogenically modified landscapes. Third, the system's robust computational architecture, built on efficient parallel processing and standardized data handling protocols, ensures scalability and reliability in processing large-scale datasets. The case studies presented demonstrate OpenBench's practical utility across various applications. The system has proven effective in identifying model strengths and areas requiring improvement, from evaluating hydrological processes and urban heat fluxes to assessing agricultural modeling capabilities. The multi-reference approach to model evaluation provides particularly valuable insights, helping distinguish between model deficiencies and observational uncertainties. OpenBench's extensible architecture ensures its continued relevance as the field evolves. The system's ability to incorporate new models, variables, datasets, and evaluation metrics allows it to adapt to emerging research needs and technological advances. This flexibility, combined with its comprehensive evaluation capabilities, positions OpenBench as a valuable resource for both model development and operational applications. Looking forward, OpenBench's role in advancing land surface modeling extends beyond technical evaluation. By providing standardized and reproducible evaluation methods, OpenBench facilitates more effective collaboration within the modeling community and supports more informed decision-making in environmental management. As we face increasing environmental challenges and seek to improve our understanding of Earth system processes, tools like OpenBench will be crucial in developing more accurate and reliable land surface models.



**Code and data availability**

All codes used in this study can be found in Wei (2024) and data used are available under a CC-BY-4.0 License at the following URL: https://doi.org/10.5281/zenodo.15064055 (Wei et al., 2025). The CoLM model used in this study can be downloaded from https://github.com/CoLM-SYSU/CoLM202X (last access: 27 March 2025). The high-resolution land surface characteristics data sets for CoLM2024 can be downloaded from http://globalchange.bnu.edu.cn/research/data. The OpenBench software is available at https://zenodo.org/records/14540647 (Wei et al., 2024) and is updated routinely at

https://github.com/CoLM-SYSU/OpenBench (last access: 26 March 2025). The TE dataset is available at https://www.eorc.jaxa.jp/water/index.html (Yamamoto et al., 2025); CLM5 dataset is available at https://rda.ucar.edu/datasets/d651011/ (Lawrence et al., 2019); GLDAS2 dataset is available at https://ldas.gsfc.nasa.gov/gldas (Rodell et al., 2004); LS3MIP dataset is available for downloading on the ESGF node https://esgf-node.llnl.gov/search/cmip6/ (Van den Hurket al., 2016); The GRDC discharge datasets are available at https://grdc.bafg.de/GRDC (The Global Runoff

Data Centre (2022)).

**Author contributions**

WZW: conceptualization and methodology of original model, software, validation, formal analysis, investigation, data curation, writing (original draft), visualization; XQC: software, validation, data curation, visualization, writing (review and editing); BF, XXH, WZX, DWZ and LHB: software, validation, data organize, writing (review and editing); WN and LXJ: resources,

method, validation data, writing (review and editing), supervision; LLB: resources, writing (review and editing), method, supervision; DYJ: resources, writing (review and editing), supervision, funding acquisition.

**Acknowledgments**

This work is supported by Southern Marine Science and Engineering Guangdong Laboratory (Zhuhai) (No. SML2023SP216), the Guangdong Major Project of Basic and Applied Basic Research (2021B0301030007) and the National Natural Science

Foundation of China (under Grants 42475172, 42088101, 42075158, 42175158, 42375166, 42077168 and 42375164). It is also supported by the National Key Scientific and Technological Infrastructure project "Earth System Science Numerical Simulator Facility" (EarthLab), and the specific research fund of The Innovation Platform for Academicians of Hainan Province (YSPTZX202143). Zhongwang Wei is supported by Guangdong Pearl River Talent Program (Young Talents) No. 2021QN02G307. We also acknowledge the high-performance computing support from the School of Atmospheric Science at

Sun Yat-sen University.



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
