# Peer review of "OpenBench: a land models evaluation system"

_EGUsphere, 2025_

## Author Comment (AC1)

**RESPONSE TO REVIEWER #1 FOR GEOSCIENTIFIC MODEL**

**DEVELOPMENT: MANUSCRIPT EGUSPHERE-2025-1380**

**BY Zhongwang Wei, Qingchen Xu, Fan Bai, Xionghui Xu, Zixin Wei, Wenzong Dong, Hongbin Liang, Nan Wei, Xingjie Lu, Lu Li, Shupeng Zhang, Hua Yuan, Laibao Liu, and Yongjiu Dai**

We thank Reviewer #1 for thoughtful and constructive feedback. This Response to the Reviewer file provides a complete documentation of the changes that have been made in response to each individual comment. Reviewer's comments are shown in plain text. Authors' responses are shown in purple color. Quotations from the revised manuscript are shown in blue color.

1. This paper presents a new software system, called OpenBench, to evaluate land surface models. OpenBench evaluates land surface models following a rigorous scientific method based on a wide range of statistical metrics and evaluation scores to allow for a quick and objective evaluation of various aspects of the models' results. OpenBench showcases its capabilities by presenting a range of analyses accompanied by a varied array of representations. Although one may deplore the general scattering of effort in the community in developing such tools, the paper is generally well written and successfully explains the advantages of the software. Using Python and well-supported packages to write the software is a solid choice, ensuring potential widespread adoption and continuous support of dependencies. The paper clearly highlights how OpenBench differs from the existing tools with support for a range of data types and of model output formats, new variables linked to human activities and the possibility of user extension for other datasets, models or variables. Although OpenBench is using common evaluation metrics and scores, the set of metrics and scores chosen is pertinent and allows for an evaluation of a wide range of aspects for land surface model results. In addition, the paper explains how OpenBench differs in its handling and visualisation of the metrics and scores.

➔ Thank you very much for your summary. I will address each of your comments and propose revisions to improve our manuscript.

2. However, a few points of the paper need to be clarified. Firstly, the choice of a Fortran namelist format for the configuration file of a Python software is unusual. Fortran namelists are not the most flexible format for configuration files and are not well supported in Python. Common, popular choices like YAML, JSON or others have a much stronger support in Python and offer greater flexibility. It would be good to explain better why the Fortran namelist format was chosen for OpenBench.

➔ Many thanks for this important suggestion. We have added support for multiple configuration formats and revised the manuscript to clarify that OpenBench now supports multiple configuration formats (YAML, JSON, and Fortran namelist), with JSON as the default choice. Please see P7L123-P7L126 in revised manuscript:

"The configuration management module accommodates three configuration namelist formats (YAML, JSON, and Fortran namelist) to meet different user preferences and workflows, with JSON as the default format. Users can utilize the configuration namelist to define evaluation parameters, data sources, and model outputs. This adaptable configuration system facilitates straightforward customization of evaluation scenarios."

3. A few points required clarification in the description of the metrics and scores. In Table 2, the bias metrics are described as "the smaller is better, ideal value is 0". However, a lot of the metrics have an infinite range [-∞,∞], in which case the smaller value for the metrics isn't 0 but -∞. It would be more accurate to say "the closer to 0 is better".

➔ Thank you for your careful check. Revised as you suggested.

4. The text explaining the various metrics used references metrics that do not appear in Table 2 and need to be clarified:

➔ Thank you for this careful observation. You are absolutely correct that there were discrepancies between the metrics described in the text and those listed in Table 2.

We have thoroughly revised Section 3.1 and Table 2 to ensure complete consistency (P12L191-P13L204):

➔ "The system incorporates various categories of metrics to capture different aspects of model performance. For example, Bias metrics, such as Percent Bias (PBIAS) and Absolute Percent Bias (APBIAS), measure systematic over- or under-estimation and bias magnitude, respectively. Error metrics, including Root Mean Square Error (RMSE), Unbiased Root Mean Square Error (ubRMSE), Centralized Root Mean Square Error (CRMSE), and Mean Absolute Error (MAE), provide different perspectives on the magnitude and nature of model errors. Efficiency metrics like Nash-Sutcliffe Efficiency (NSE) and Kling-Gupta Efficiency (KGE) evaluate model performance relative to baselines and combine multiple aspects of the model-data agreement. Correlation metrics, including Pearson correlation coefficient (R) and coefficient of determination ($R^2$), quantify the strength and direction of linear relationships between model outputs and observations. The Index of Agreement (IA) provides a more comprehensive assessment of magnitude and phase agreement. Variability metrics such as Ratio of Standard Deviations (rSD) and specialized bias metrics for maximum (PC_MAX), minimum (PC_MIN), and amplitude (PC_AMPLI) values help identify whether models accurately capture the range of system variability and extreme conditions. For categorical data, the Cohen's Kappa coefficient (KC) evaluates agreement while accounting for chance. Variability metrics such as Relative Variability (RV) and Coefficient of Variation (CV) help identify whether models accurately capture the range of system variability."

5. line 176: "For categorical data, the **Kappa** coefficient", (bolding from me)

➔ Thank you for identifying this specific inconsistency. We have revised Table 2 to include the Kappa coefficient (P13L201-P13L202).

"For categorical data, the Cohen's Kappa coefficient (KC) evaluates agreement while accounting for chance."

6. line 178: "and **Percent Change** in maximum and minimum values help identify" (bolding from me)

➔ Thanks. We have corrected it (P13L202-P13L204):

"Variability metrics such as Relative Variability (RV) and Coefficient of Variation (CV) help identify whether models accurately capture the range of system variability."

7. The variable naming in the calculation of nRMSEScore should be reviewed. The name "CRESM" is strange; shouldn't it be "CRMSE"? Additionally, the error is once called εrmse and once εcresm.

➔ Thank you very much for careful check. This is a typo; it should be CRMSE. We have revised it.

8. The nPhaseScore score explanation needs to be reviewed. Several issues with it are likely linked and can be addressed together. I do not understand what "climatological mean cycles" (line 214) are, which cycles are referred to here? I also do not understand what is referred to with "of evaluation time resolution". Finally, there are two mathematical symbols in the equations that are not explained, λ and φ.

➔ Thank you very much. You are correct that several terms require better definition. We have revised our expression (P14L229P15L237):

"The nPhaseScore is calculated as:

$$\text{nPhaseScore}(x) = \frac{1}{2}\left[1 + cos\left(\frac{2\pi\theta(x)}{nstep}\right)\right] \tag{5}$$

where $\theta(x, \lambda, \phi)$ is the time difference between modeled and observed maxima:

$$\theta(x) = maxima\left(c_{sim}(x, t)\right) - maxima\left(c_{ref}(x, t)\right) \tag{6}$$

Here, $c_{sim}$ and $c_{ref}$ are the climatological mean cycles (i.e., the average seasonal patterns) of the model and reference data, computed by averaging each month or day across all years in the time series. The $"maxima"$ ( ) identifies the timing (month for monthly data, day for daily data) when the peak value occurs in these average seasonal cycles at each spatial location x. The parameter *nstep* represents the number

of time steps in a complete annual cycle (e.g., 12 for monthly data or 365 for daily data) and normalizes the phase difference to the annual cycle."

➜ The symbols λ (longitude) and φ (latitude) have been removed from the equations as they were redundant with the spatial coordinate x.

9. Lastly, there is very little explanation of the nSpatialScore score. Why was this done so?

➜ Thank you very much. We apologize for this oversight. We have added it (P15L246-P16L252):

"The Spatial Score (nSpatialScore) evaluates how well the model captures the spatial distribution of a variable compared to observations by assessing both the spatial correlation and the relative variability across the domain. The nSpatialScore is calculated as:

$$\text{nSpatialScore} = 2(1 + R)\Big/\left(\sigma + \frac{1}{\sigma}\right)^2 \tag{10}$$

where R is the spatial correlation coefficient between the model and reference period mean values, and σ is the ratio of spatial standard deviations:

$$\sigma = stdev\left(\overline{v_{sim}}(x)\right)\Big/stdev\left(\overline{v_{ref}}(x)\right) \tag{11}$$

"

10. In the section showcasing the tool with some use cases, I disagree with the conclusion of the urban heat evaluation that "these findings highlight the importance of refined urban parameterization schemes in land surface models" (line 377). It isn't clear why the results shown indicate this. The results indicate the CoLM2024 model performs well except for a specific zone, but do not show if models with different parameterizations do better or worse. Although I strongly agree that refined urban parametrizations can perform better, I disagree that the

results shown in this paper allow us to draw a conclusion on the importance of urban parametrization.

➔ Thank you for this important correction. We totally agree with your opinion. We have revised this part (P22L406-P22L408):

"While the exact mechanisms driving these regional differences are still being investigated, these results demonstrate OpenBench's ability to identify spatial patterns of model-observation disagreement that require further exploration."

11. In the multiple models comparison, at line 456, it says CoLM2024 and TE are the best models for canopy transpiration and total runoff, whereas figure 6 shows CLM5 and CoLM2024 are the best for the total runoff. The text in this section talks of "superior performance". I would argue that we can't qualify a score of 0.54 for the runoff of superior. It seems "highest" might be a better choice of qualifier in this case.

➔ Thank you very much. We have revised the related content (P28L487-P28L489):

"The analysis reveals that under current configurations, CoLM2024 and TE achieve the highest score for canopy transpiration, while CLM5 and CoLM2024 show the highest score for total runoff. CoLM2024 maintains relatively higher score in other variables."

12. In the multiple models comparison section, I also question the choice of the vertical axis range in the parallel coordinates plot for the scores (figures 6b and 7b). I think these plots would be more informative if OpenBench used the same range from 0 to 1 for all the plots. In this way, the plots would visually highlight not only the relative position of the various models, but also the overall quality of all the models (how far from 1 all the models lie) and the relative performance of the models between each other (the spread of the lines would visually highlight if the models performed similarly or very differently). It would make it harder to identify small differences between models, which is, in my view, an advantage as small differences indicate similar performances. It is logical to keep the setting of the range for the vertical axis unchanged in the parallel coordinates plot for the metrics since, contrary to scores, a lot of metrics have an infinite range.

➔ Thank you very much. We appreciate the suggestion to standardize the vertical axis range for the score plots in Figures 6b and 7b. We agree that this adjustment better visualizes both the absolute performance and relative differences between models, while deemphasizing minor variations. As requested, we have modified these figures to use same range for scores (attached below).

[Figure]

[Figure]

13. In the section comparing a model to multiple reference datasets, I find Figure 9 confusing. It presents a heatmap of various metrics for a model compared to several datasets. The same colormap is used for all metrics, with darker hues for higher metric values. Unfortunately, the metrics do not all show a better agreement at the higher values. Users then need to know the details of each of the metrics to interpret the table instead of being visually guided by the figure. This representation of the metrics would work better if OpenBench used different colormaps for different types of metrics: closer to 0 metrics with a darker hue at 0, metrics with the smallest values being the best with a darker hue for the smallest values, etc. I realise it is harder to put together, but it would greatly improve the representation.

➔ Thank you for your insightful observation regarding the interpretability of Figure 9. We agree that using a single colormap for all metrics, despite their differing interpretations, could mislead readers without prior knowledge of each metric's directionality. To address this, we have revised the figure by implementing metric-specific colormaps.

[Figure]

14. Finally, the paper refers several times to the efficiency of the tool and points out the parallelisation using Dask. However, there is nothing in the paper to substantiate this. It would be good if some information could be given about the resources used and the time needed to produce the analyses that are showcased in the paper, for example.

➔ Thanks for the valuable suggestion. We have added a subsection to show the efficiency of OpenBench (P8L163-P9L183):

"OpenBench showcases remarkable efficiency benefits thanks to its parallel processing architecture. Assessment results from typical workloads reveal considerable advancements compared to sequential processing methods. In station-based evaluations, which involve IO-intensive tasks due to the requirement to read and process multiple

individual site files, OpenBench demonstrates outstanding scalability. Evaluating a single variable across 142 stations takes about 3.12 minutes when using single-process execution. However, with parallel processing utilizing 48 cores, this time is reduced to only 0.509 minutes on an Intel(R) Xeon(R) CPU E5-4640 v4 @ 2.10GHz with 48GB RAM, thanks to Joblib's effective task distribution. For gridded data processing, OpenBench employs Dask's lazy execution and chunked array processing to effectively manage memory while ensuring high processing speeds. Processing model outputs at a 0.25° resolution from 2001 to 2010 with monthly temporal resolution against two reference datasets takes approximately 2.302 minutes with sequential processing, but only 1.301 minutes when leveraging Dask's parallel capabilities on the same hardware configuration. These performance improvements are particularly beneficial for thorough model evaluations that involve multiple variables, reference datasets, and spatial domains. The impressive scalability with available cores makes OpenBench ideal for both rapid diagnostic evaluations on personal workstations and extensive comparative studies on high-performance computing systems. Additionally, effective memory management guarantees that analyses can be conducted even on systems with limited memory allocations, thus enabling high-resolution, in-depth model evaluation capabilities.

The current test data shows notable performance improvements, but the advantages of OpenBench's parallel processing architecture stand out even more with higher resolution datasets and temporal resolution in gridded data processing. Here, the efficiency gains from Dask's parallel capabilities grow more substantial. In station-based evaluations, the improvement in performance scales with the number of stations being assessed. As the number of stations rises, the decrease in processing time through parallel execution using additional cores becomes significantly greater, further emphasizing OpenBench's scalability for extensive, high-resolution analyses. "

**Technical corrections:**

Bold text indicates parts of the cited text that I modified to show needed corrections.

15. Line 29 and 31: "various changes in the Earth system", "key components of Earth system models". "Earth", when referring to the planet, takes an uppercase

➔ Thank you very much. We have corrected it (P1L30-P1L32).

"As such, they are key components of Earth system models (ESMs) and have significant impacts on our ability to comprehend and predict weather, climate, hydrological cycles, carbon cycles, and various other environmental factors."

16. Line 164: "For example, bias metrics", no uppercase to "bias".

➔ Thank you for this grammatical correction. Corrected (P12L191-P12L193).

"For example, Bias metrics, such as Percent Bias (PBIAS) and Absolute Percent Bias (APBIAS), measure systematic over- or under-estimation and bias magnitude, respectively."

17. Line 188: "For a given variable $v(t, x)$, where $t$ represents time and $x$ represents spatial coordinates, we first calculate". The first sentence here is not a sentence; replace the full stop after "coordinates" with a comma.

➔ Corrected. Thanks (P13L213-P14L215).

"For a given variable $v(t, x)$, where $t$ represents time and $x$ represents spatial coordinates, we first calculate the bias from the temporal means of both the reference $\overline{v_{ref}}(x)$ and model $\overline{v_{sim}}(x)$ data."

18. Line 194: "Where t0 and tf are the first and final timesteps, respectively." Replace singular with plural.

➔ Thank you for this correction. Corrected (P14L218-P14L219).

"Where $t_0$ and $t_f$ are the first and final timestep, respectively. We then compute the bias, $bias(x) = v_{ref}(t, x) - v_{sim}(t, x)$. The relative error in bias is then given as $\varepsilon_{bias}(x) = \frac{|bias(x)|}{CRMS(x)}$. The bias score as a function of space is then computed as:"

19. Line 200: "Similarly to nBiasScore, we first calculate the centralized RMSE:". "Similar" changed to "Similarly", "We" changed to "we", "RSME" changed to "RMSE", and remove bolding of nBiasScore.

➜ Thank you very much. Revised as you suggested (P14L222-P14L223).

"Similar to nBiasScore, we first calculate the Centralized RMSE (CRMSE)"

20. Line 239: "In contrast, OpenBench offers". Replace "offering" with "offers".

➜ Thank you very much. Revised as you suggested (P16L263-P16L263).

"In contrast, OpenBench offers greater flexibility in its weighting methods."

21. Line 277: The sentence finishing with "making it possible to evaluate." is incomplete. It should be combined with the next sentence.

➜ Thank you for the careful check. Revise as you suggested (P17L304-P17L306).

"The resolution ranges from coarse (e.g., 0.5° for ILAMB datasets (Collier et al., 2018)) to very fine (e.g., 500m for MODIS-based products (Varquez et al., 2021)), making it possible to evaluate LSMs across different spatial scales, from global assessments to regional or plot-scale studies."

22. Figure 3 legend: Replace with "An example of a scores heatmap for GPP classified by IGBP land cover."

➜ Thank you. Revised as you suggested (P22L309-P22L400).

[Figure]

Figure 3: An example of a scores heatmap for GPP classified by IGBP land cover

23. Line 293: Considering OpenBench does not provide any datasets, the part saying "while OpenBench integrates a comprehensive collection of datasets," would be more accurate as such: "while OpenBench integrates with a comprehensive collection of datasets,"

➔ Thank you. Revised as you suggested (P18L321-P18L322).

"It is noted that while OpenBench integrates with a comprehensive collection of datasets, we cannot directly provide specific data due to copyright restrictions and licensing agreements."

Citation: https://doi.org/10.5194/egusphere-2025-1380-RC1

---

## Author Comment (AC4)

**RESPONSE TO REVIEWER #2 FOR GEOSCIENTIFIC MODEL DEVELOPMENT: MANUSCRIPT EGUSPHERE-2025-1380**

BY Zhongwang Wei, Qingchen Xu, Fan Bai, Xionghui Xu, Zixin Wei, Wenzong Dong, Hongbin Liang, Nan Wei, Xingjie Lu, Lu Li, Shupeng Zhang, Hua Yuan, Laibao Liu, and Yongjiu Dai

We thank Reviewer #2 for thoughtful and constructive feedback. This Response to the Reviewer file provides complete documentation of the changes that have been made in response to each individual comment. Reviewer's comments are shown in plain text. Authors' responses are shown in purple. Quotations from the revised manuscript are shown in blue.

1. This paper describes new cross-platform software system for evaluation and comparison of land surface models using a broad suite of metrics, statistics and comparison methods. Authors clearly demonstrate OpenBench's capabilities with various examples. Figures are comprehensive and clear. The manuscript is written very clearly, with few grammatical errors, and therefore I have few comments in this regard.

➔ Thank you very much for this positive assessment of our manuscript. We greatly appreciate your recognition. I will address each of your comments and propose revisions to improve our manuscript.

2. Regarding the software itself, I appreciate authors efforts to provide an easily accessible and runnable code base along with sample data for testing. However, I note if users follow "usage" instructions from the github repository README, there is no file provided for "nml/main.nml", so the program fails. I was able to run the more complex example with sample data using the file "main-Debug.nml", but I recommend authors update the codebase to provide a highly simplified "main.nml" for initial user testing, and clearer instructions on how to adapt the codebase for custom models/dataset analysis.

➔ Thank you for this valuable feedback. We have thoroughly revised the README of the GitHub repository to provide an accurate overview and clear step-by-step instructions for users. Additionally, we have developed and included a comprehensive user manual (located in the doc folder) that contains complete installation instructions, step-by-step tutorials from basic to advanced usage, clear examples demonstrating how to adapt OpenBench for custom models and datasets, and troubleshooting guidance for common issues. We have also ensured that all referenced configuration files, including a test namelist for initial testing, are properly included in the repository with clear documentation of their purpose and usage.

3. An internet connection is required for some plotting functions (e.g. to download Cartopy coastline), while some HPC environments may not have internet connectivity. Without internet connectivity, the program fails. A programmed exception to exclude downloading coastlines etc would improve functionality.

➔ Thank you very much. This is indeed a common constraint in many institutional computing systems. Since Cartopy is the main package for our plotting functions and Basemap is no longer actively maintained, we cannot exclude Cartopy without significantly compromising the visualization capabilities that are central to OpenBench's functionality. To address this issue, we have incorporated comprehensive troubleshooting guidance into readme and our user manual, which offers detailed instructions on how to manually download and install Cartopy map files in offline environments. This includes step-by-step procedures for pre-downloading the requisite coastline and boundary data on internet-connected systems and subsequently transferring them to HPC environments, as well as providing configuration instructions for directing Cartopy to utilize these local data files.

4. Regarding the manuscript, authors may wish to comment in the paper on the name "OpenBench", and reduce reference to this being a "benchmarking system", as readers may have a different interpretation of "benchmarking". To my understanding, the broad meaning of benchmarking is comparison with a welldefined standard, or an a-priori performance expectation (e.g. see introduction and explanatory figures in your reference Best et al., 2015). This software undertakes evaluation and comparison without explicitly benchmarking (using the definitions in Best et al.,). However, I recognise that others in the community use "benchmarking" differently (e.g. in ILAMB). This could be commented on in the paper.

➔ Thank you for the excellent point regarding the conceptual distinction between "benchmarking" and "evaluation". We agree that the term "benchmarking" can have different interpretations within the modeling community. In the strictest sense, benchmarking implies comparison against well-defined performance standards or a priori expectations, as described by Best et al. (2015). However, following the precedent established by community tools such as ILAMB (International Land Model Benchmarking), we use "benchmarking" in the broader sense of systematic model evaluation and comparison against observational datasets, without necessarily establishing predetermined performance thresholds. We have added the following content to address this issue (P16L280-P17L284):

"It is worth noting that, although we refer to OpenBench as a "benchmarking system" in accordance with community convention, the tool primarily functions as an evaluation and comparison framework rather than adhering to strict benchmarking with predetermined performance standards. This design choice affords users the flexibility to establish their own performance criteria while benefiting from standardized evaluation methodologies."

5. Some referenced models, datasets or studies are not properly referenced. For example: CLASS, CABLE, PLUMBER2. Please include relevant references.

➔ Thank you very much. We have included relevant references.

6. Also ensure all acronyms are defined. For example, I cannot find a definition for uRMSD used in Figure 10. Overall, figure captions could be improved by reducing or explaining acronyms.

➔ We sincerely appreciate the reviewer's careful attention to terminology accuracy. In the revised manuscript, we have made the correction as below: The originally

labeled "uRMSD" has been corrected to "ubRMSE", and "RMSD" has been updated to "RMSE". All acronyms throughout the manuscript have been verified and properly defined. Figure captions have been reviewed to ensure technical terms are either spelled out or properly referenced to their definitions in the text. These changes improve accessibility for readers and maintain consistency across the manuscript. Thank you for highlighting this issue.

[Figure]

7. Please ensure software in Table 2 is properly named. For example ESMVal should be ESMValTool, and PALS has changed their name to modelevaluation.org.

➔ Thank you very much. Corrected. We have also reviewed all software names in Table 2 to ensure they are properly named.

8. Overall, I see great potential to this work, and congratulate authors for this contribution. I look forward to integrating OpenBench into my evaluation workflow.

➔ Many thanks. We are delighted to hear that you plan to integrate OpenBench into your evaluation workflow. Please feel free to contact us via our GitHub repository or reach out directly if you encounter any issues or have suggestions for improvements.

---

## Author Response (AR2)

**RESPONSE TO REVIEWER #1 FOR GEOSCIENTIFIC MODEL DEVELOPMENT: MANUSCRIPT EGUSPHERE-2025-1380**

**BY Zhongwang Wei, Qingchen Xu, Fan Bai, Xionghui Xu, Zixin Wei, Wenzong Dong, Hongbin Liang, Nan Wei, Xingjie Lu, Lu Li, Shupeng Zhang, Hua Yuan, Laibao Liu, and Yongjiu Dai**

We thank Reviewer #1 for thoughtful and constructive feedback. This Response to the Reviewer file provides a complete documentation of the changes that have been made in response to each individual comment. Reviewer's comments are shown in plain text. Authors' responses and quotations from the revised manuscript are shown after **Reply**.

1.  I thank authors for providing this revised manuscript, and for incorporating previous suggestions. I have two suggestions relating to this latest revision.

**Reply:**

Thank you very much for your review. I will address each of your comments and propose revisions to improve our manuscript.

2.  Authors introduce a discussion on the parallelisation of the software, describing its performance as demonstrating "remarkable efficiency" and "outstanding scalability." However, the results comparing 1 to 48 cores indicate that the scalability falls significantly short of linear. I recommend moderating this language to simply state that parallelisation provides speed improvements. Additionally, the section could be made more concise, as it is quite long.

**Reply:**

Thanks for the valuable suggestion. We have revised our manuscript (**P8L163-P8L173**):

**P8L163-P8L173:** "OpenBench achieves speed improvements through its parallel processing architecture. Benchmark tests demonstrate clear advantages over sequential processing methods. In station-based evaluations, processing a single variable across 142 stations takes 3.12 minutes using single-process execution, whereas parallel processing with 48 cores reduces this to 0.509 minutes on an Intel(R) Xeon(R) CPU E5-4640 v4 @ 2.10 GHz with 48 GB RAM. OpenBench uses Dask's lazy execution

and chunked arrays for efficient gridded data processing, balancing memory use and processing speed. Processing 0.25° resolution model outputs (2001-2010, monthly) against two reference datasets takes 2.302 minutes sequentially versus 1.301 minutes in parallel on the same hardware. These performance improvements are particularly beneficial for comprehensive model evaluations involving multiple variables, reference datasets, and spatial domains. The efficiency gains from parallel processing become more substantial with higher-resolution datasets and increasing numbers of evaluation sites, making OpenBench suitable for both rapid diagnostic evaluations on personal workstations and extensive comparative studies on high-performance computing systems."

3. Authors have linked to an updated software repository, and a thoroughly revised README here: https://github.com/zhongwangwei/OpenBench. I tried running the updated software following "Usage" instructions, with the command:

python script/openbench.py nml/main-Debug.json.

Unfortunately, when running on my HPC I encountered this error:

FileNotFoundError: [Errno 2] No such file or directory: './output/Debug/output/scores/Evapotranspiration_stn_GLEAM_hybird_PLUMBER2 _station_case_evaluations.csv'

In my first review I was able to run this to completion without error. I also suggested in my first review that authors provide a stripped down, highly simplified example script which will allow new users to test a basic version of OpenBench on their system (the previous example test was "main.json"). I reiterate this suggestion, as "main-Debug.json" appears to be a complex configuration for debugging purposes, i.e. it runs more than a simple case.

**Reply:**

We appreciate the reviewer's rigorous testing and valuable feedback. We have ffixed the reported error. Please find our responses below:

1) The reported FileNotFoundError has been fixed. We recently developed an "only_drawing" module to support secondary visualization of existing evaluation

results. The error occurred because this option was mistakenly set to true in the configuration file. We have corrected this by setting it to false in the default configuration.

2) We continue to recommend using main-Debug.json as the primary example script, as it enables comprehensive testing of all four evaluation types: Grid-Grid, Grid-Station, Station-Grid, and Station-Station.

3) Users can selectively disable specific tests by modifying the configuration file main-Debug.json according to their testing needs.

We sincerely appreciate the reviewer's suggestions that helped improve OpenBench's accessibility. Please don't hesitate to contact us if further issues arise.